# PROTOBIND-DIFF: PROTEIN-CONDITIONED DISCRETE DIFFUSION FOR STRUCTURE-FREE LIGAND GENERATION

## ABSTRACT

Designing small molecules that selectively bind to protein targets remains a central challenge in drug discovery. While recent generative models leverage 3D structural data to guide ligand generation, their applicability is limited by the sparsity and bias of experimentally determined complexes. Here, we introduce ProtoBind-Diff, a *structure-free masked diffusion model* that conditions molecular generation directly on protein sequences via pre-trained language model embeddings. Trained *on over one million active protein-ligand pairs from BindingDB*, ProtoBind-Diff generates chemically valid, novel, and target-specific ligands without requiring 3D structures for inference. In extensive benchmarking against 3D structure-based models, ProtoBind-Diff achieves competitive predicted binding affinity scores and performs well on challenging targets, including those with limited training data. Despite never being trained on the data that contain binding pockets, its attention maps align with contact residues, suggesting the model learns spatially meaningful interaction priors from sequence alone. These results demonstrate that sequence-conditioned diffusion can enable *structure-free, scalable ligand discovery across the proteome*, including orphan or rapidly emerging targets.

## 1 INTRODUCTION

The chemical space of drug-like molecules is estimated to exceed $10^{60}$ structures [Polishchuk et al. (2013)], making exhaustive exploration practically infeasible. Machine learning has emerged as a powerful tool to generate candidate compounds, guiding discovery beyond what traditional screening can reach. Generative AI models have been developed to address this challenge, leveraging various molecular representations, such as text strings, graphs, or 3D structures, and spanning a wide range of architectures, including transformers [Chithrananda et al. (2020b); Bagal et al. (2021)], reinforcement learning agents [Loeffler et al. (2024a)], variational autoencoders (VAEs) and generative adversarial networks (GANs) [Simonovsky & Komodakis (2018); De Cao & Kipf (2018)], and more recently, diffusion models [Jo et al. (2024); Vignac et al. (2023)].

A promising yet challenging frontier is *protein-conditioned molecular generation*, where models design ligands specific to a biological target. Recent approaches have focused on using 3D structures of protein-ligand complexes or binding pockets (e.g., DiffDock [Corso et al. (2022)], EquiBind [Stärk et al. (2022)], TargetDiff [Guan et al. (2023)]) to either predict optimal docking poses for given molecules or generate novel molecules directly within binding sites. However, these models face several critical limitations. First, many methods assume static binding sites and overlook conformational flexibility and induced-fit effects, which are often essential for ligand potency. Second, they rely on paired protein-ligand structural data, which remains limited (fewer than 30,000 complexes in the PDBbind database [Liu et al. (2015)]) and biased towards well-studied targets and chemotypes. Third, structure-based optimization can constrain chemical diversity, prioritizing docking fit over meaningful properties such as drug-likeness, pharmacokinetics, or novelty.

It is important to emphasize that several recent approaches explicitly incorporate protein pocket flexibility by jointly generating ligand and holo-like pocket conformations from apo structures [Zhou et al. (2025); Zhang et al. (2024b)]. These methods directly model conformational changes.

In this work, we propose ProtoBind-Diff, a *masked diffusion language model* for molecular generation conditioned on protein sequence, *bypassing the need for 3D structures for training*. To develop ProtoBind-Diff, we frame molecular generation as a denoising process over the vocabulary of molecular tokens by incorporating recent works on discrete diffusion [Sahoo et al. (2024)]. We propose a mechanism for directed protein-specific generation of molecules by adding *condition of pre-trained embeddings of protein sequence via cross-attention block*. To improve robustness and diversity, we propose cluster-based resampling of training molecules and add token permutation augmentation. Avoiding the need for 3D structures enabled us to train the model on over one million active protein-ligand pairs from BindingDB [Gilson et al. (2016)], a scale far exceeding what is feasible with structure-based datasets such as PDBbind.

We further demonstrate that ProtoBind-Diff generates molecules that preserve physicochemical properties of known actives, achieves competitive or superior affinity metrics over structure-based baselines (Pocket2Mol, PocketFlow), and performs well to low-data targets. To compare the performance of different models, a comprehensive benchmark consisting of 12 protein targets was constructed consisting of both frequently and infrequently represented proteins in classical training datasets (PDBBind and BindingDB). On this benchmark, results further demonstrate that Boltz-1 constitutes more objective and discriminative evaluation metrics than Vina docking. Attention analysis shows that *cross-attention heads consistently highlight binding site residues*, suggesting that the model encodes biophysically meaningful interaction patterns.

To summarize, the main contributions of this work are:

- We propose a masked discrete diffusion framework for target-aware molecular generation that conditions on protein sequence embeddings via cross-attention.
- We propose a dataset resampling scheme that increases the diversity of generated molecules by clustering similar molecular structures.
- We evaluate on a 12-target benchmark comprising both frequently and sparsely represented proteins and find that Boltz-1 is more reliable and discriminative than classical docking in our experiments.
- We show improved molecular quality: our model achieves higher enrichment based on Boltz-1 evaluation and yields molecular property distributions closer to actives than baselines.
- We demonstrate that specific cross-attention heads focus on binding site residues, offering biologically grounded interpretability.

## 2 RELATED WORK

**Diffusion Language Models.** Diffusion models have recently emerged as strong alternatives to autoregressive methods for discrete data generation [Nie et al. (2025); He et al. (2023)]. Unlike autoregressive approaches, discrete diffusion enables parallel sampling and bidirectional context, which is especially important for generating chemically valid and diverse molecules. In [Austin et al. (2021)], diffusion probabilistic models were extended to discrete categorical data by defining a forward Markov corruption process and the corresponding ELBO for likelihood-based training. Lou et al. (2024) proposed a score-based variant that maps discrete tokens to a continuous space, at the expense of an explicit likelihood. Sahoo et al. (2024) simplified masked diffusion training through a Rao-Blackwellized ELBO, reducing to mixtures of MLM losses, while Shi et al. (2024) introduced a continuous-time formulation unifying different modalities. These advances enabled first applications to molecular strings, e.g., GenMol [Lee et al. (2025)] and PepTune [Tang et al. (2025)], which demonstrated that masked discrete diffusion can learn molecular syntax and generate valid compounds. To our knowledge, no prior work integrates protein embeddings as conditioning inputs into masked discrete diffusion for target-aware molecule generation, which is the focus of this paper.

**Textual Molecular Representation.** SMILES strings [Weininger (1988)] remain the dominant sequence representation due to their simplicity and compatibility with language models, and have powered large-scale pretraining [Irwin et al. (2022); Chithrananda et al. (2020a); Lu & Zhang (2022). Their main drawback is fragility: small perturbations can render molecules invalid. SELF-IES [Krenn et al. (2020)] guarantee validity but sacrifice simplicity and interpretability. Comparative

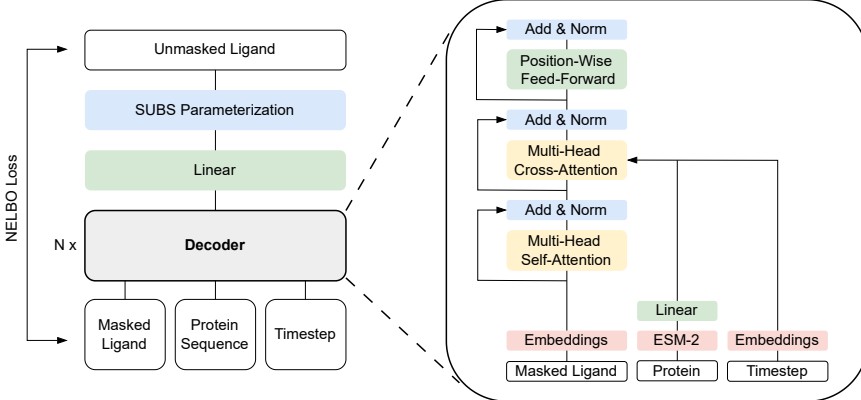

Figure 1: Architecture of the ProtoBind-Diff model. The masked ligand sequence is embedded, then processed through a stack of transformer decoder blocks. Each block contains multi-head self-attention with rotary position embeddings, multi-head cross-attention for protein sequence and timestep conditioning, followed by a normalization layer and a position-wise feed-forward network. Protein sequence information is encoded using pre-trained ESM-2 embeddings and projected through a linear layer. The final output is passed through a linear layer and SUBS parameterization to predict the denoised ligand.

studies [Chithrananda et al. (2020a); Gao et al. (2022); Leon et al. (2024)] report mixed results, with SMILES often outperforming SELFIES in practice. Recently, SAFE [Noutahi et al. (2024)] introduced a fragment-based representation tailored for scaffold decoration and linking. Several masked diffusion models [Tang et al. (2025); Wang et al. (2025); Lee et al. (2025)] also adopt SMILES or SAFE, reflecting their robustness and flexibility. In this work, we follow this trend and employ SMILES with augmentation to improve model generalization.

**Context-dependent Molecular Generation.** Context-aware molecular generation aims to design ligands that bind to a specific protein. Protein binding inherently happens in 3D space, so many models leverage structural information to incorporate protein context. For example, Xu et al. (2021) introduced a cRNN conditioned on pocket descriptors. Ragoza et al. (2022) proposed one of the first 3D molecule generators, a conditional VAE that encodes receptor-ligand complexes as 3D atomic density grids and decodes new ligand density maps, from which discrete molecules are reconstructed. Li et al. (2021) presented DeepLigBuilder, combining a 3D graph generator with Monte-Carlo Tree Search to design ligands inside protein pockets. Autoregressive 3D methods place atoms sequentially in a pocket with GNNs (e.g., GraphBP [Liu et al. (2022)] uses local frames and a flow head), while Pocket2Mol [Peng et al. (2022)] introduces an E(3)-equivariant pocket encoder and an efficient conditional sampler that assembles ligands inside 3D pockets, modelling both geometry and bonding. The field is now dominated by 3D pocket-based diffusion: TargetDiff [Guan et al. (2023)] jointly denoises coordinates and atom types with an SE(3)-equivariant network and also provides unsupervised affinity features for ranking, whereas DiffSBDD [Schneuing et al. (2024)] frames SBDD as an SE(3)-equivariant conditional diffusion process that enables joint 3D ligand generation with support for constraint-guided inpainting and direct structure optimization. Complementary directions include PocketFlow [Jiang et al. (2024)], an autoregressive flow model generating 3D ligands inside protein pockets using chemical constraints, confirmed by wet-lab validated bioactive hits, and TamGen [Wu et al. (2024)], a GPT-like chemical language model for target-aware SMILES generation and compound refinement. Recent approaches such as FlexSBDD [Zhang et al. (2024b)] and DynamicFlow [Zhou et al. (2025)] explicitly model induced-fit effects by jointly updating protein degrees of freedom and ligand poses from apo structures. Despite this progress, most of these approaches rely on relatively scarce 3D training data, and remain limited to targets with well-characterized binding pockets and reliable structural information.

## 3 METHODS

We trained and evaluated ProtoBind-Diff, a structure-free, protein sequence-conditioned masked diffusion model for molecular generation. Protein context is provided through ESM-2 [Lin et al. (2022)] embeddings, which are integrated via cross-attention to guide the reconstruction of masked tokens in a SMILES string.

### 3.1 MASKED DISCRETE DIFFUSION

We employ a masked discrete diffusion framework to generate SMILES sequences conditioned on the protein target. Protobind-Diff follows the MDLM training paradigm with a transformer decoder [Vaswani et al. (2017)] backbone, enhanced by timestep and protein-sequence conditioning, and also rotary positional embeddings [Su et al. (2024)]. Our choice of discrete diffusion over autoregressive modeling is motivated by two factors: (i) the superior computational efficiency of non-autoregressive parallel decoding, and (ii) the ability of bidirectional attention to exploit molecular context without depending on a fixed token ordering, which aligns more naturally with molecular representations.

We define masked diffusion in line with MDLM [Sahoo et al. (2024)]. A ligand is represented as a sequence of tokens $\mathbf{x} = (\mathbf{x}^1, \dots, \mathbf{x}^L)$, where each token $\mathbf{x}_i^l \in \{0, 1\}^K$ is a one-hot vector over $K$ categories (with $\sum_{i=1}^{K} \mathbf{x}_i^l = 1$). We define a categorical distribution $\mathrm{Cat}\left(\cdot; \boldsymbol{\pi}\right)$ over $K$ classes (with probabilities given by $\boldsymbol{\pi} \in \Delta^K$), where $\Delta^K$ represents the simplex over $K$ categories. We assume the $K$-th category serves as the masking token, with one-hot vector $\mathbf{m}$, i.e., $\mathbf{m}_K = 1$.

The forward process $q$ interpolates between each token in clean data sequence $\mathbf{x}^l$ and a target distribution $\mathrm{Cat}\left(\cdot; \mathbf{m}\right)$ (in case of masked diffusion we set $\boldsymbol{\pi} = \mathbf{m}$), and is defined as:

$$q\left(\mathbf{z}_t^l \mid \mathbf{x}^l\right) = \mathrm{Cat}\left(\mathbf{z}_t^l; \alpha_t \mathbf{x}^l + (1 - \alpha_t)\mathbf{m}\right), \qquad (1)$$

where $\mathbf{z}_t^l$ denotes the $l$-th token in the noisy sequence at time $t$, with $t$ ranging from $t = 0$ (clean) to $t = 1$ (most noisy). The masking ratio $\alpha_t \in [0, 1]$ is a strictly decreasing function of $t$, with $\alpha_0 \approx 1$ and $\alpha_1 \approx 0$.

The reverse unmasking process inverts the forward noise process $q$. An optimal form for the posterior of the reverse process matches the true posterior:

$$p_\theta(\mathbf{z}_s^l \mid \mathbf{z}_t^l) = q(\mathbf{z}_s^l \mid \mathbf{z}_t^l, \mathbf{x}^l) = \begin{cases} \mathrm{Cat}(\mathbf{z}_s^l; \mathbf{z}_t^l), & \mathbf{z}_t^l \neq \mathbf{m}, \\ \mathrm{Cat}\left(\mathbf{z}_s^l; \dfrac{(1 - \alpha_s)\mathbf{m} + (\alpha_s - \alpha_t)\mathbf{x}^l}{1 - \alpha_t}\right), & \mathbf{z}_t^l = \mathbf{m}. \end{cases} \qquad (2)$$

where step $s < t$. The posterior is conditioned on unknown $\mathbf{x}^l$, so different parameterization techniques can be used to approximate $\mathbf{x}$ with a neural network $\mathbf{x}_\theta(\mathbf{z}_t, t)$. We used the substitution-based (SUBS) parameterization approach described in Sahoo et al. (2024). In this parameterization design, the unmasked tokens remain unchanged during the reverse diffusion, and the clean input is not masked. Assuming that the forward noise process is applied independently throughout the sequence, the training objective of $\mathbf{x}_\theta$, approximated by the negative ELBO, is formulated as

$$\mathcal{L}_{\mathrm{NELBO}} = \mathbb{E}_q \int_0^1 \frac{\alpha_t'}{1 - \alpha_t} \sum_{l=1}^{L} \log \left\langle \mathbf{x}_\theta^l\left(\mathbf{z}_t^{1:L}, t\right), \mathbf{x}^l \right\rangle \, \mathrm{d}t, \qquad (3)$$

where $\mathbf{x}_{\boldsymbol{\theta}}^l$ is a predicted value of $l$-th token, $l = \overline{1, L}$. This objective is a weighted average of masked language modeling (MLM) losses across diffusion timesteps.

### 3.2 MODEL ARCHITECTURE

At each timestep $t$, the model receives the masked ligand sequence together with an embedded protein sequence and a timestep embedding, and predicts per-position logits over a vocabulary of size $K$. We adopt the log-linear noise schedule proposed by Sahoo et al. (2024). As the backbone, we use a Transformer decoder with a cross-attention layer for conditioning Vaswani et al. (2017) (Figure 1). After the ligand token sequence is embedded, rotary positional embeddings are applied in the self-attention layer [Su et al. (2024)]. Protein features are obtained from a frozen ESM-2

model by taking the last hidden-layer representations and projecting them with a linear layer to the decoder's hidden size [Lin et al. (2022)]. We concatenate the timestep embedding with the projected protein embeddings and pass the result to the cross-attention layer. The final linear layer maps the decoder outputs to logits over the vocabulary, after which we apply the SUBS parameterization. SUBS ensures the network denoises only masked tokens by passing the logits at unmasked positions and setting the logit for the mask token $\mathbf{m}$ to $-\infty$. Although MDLM originally used a diffusion Transformer with adaptive layer normalization for timestep conditioning [Peebles & Xie (2022)], we adopt the architecture above to condition on the full protein sequence. Moreover, the cross-attention layer directly models interactions between protein and ligand tokens and enables interpretation via attention maps (see Attention-Based Binding Site Analysis).

To improve generalization and reduce overfitting, we use SMILES augmentation, randomizing strings while preserving chemical validity as described by Arús-Pous et al. (2019). To reduce redundancy, we cluster highly similar molecules in the training set and sample one representative per cluster at each epoch, reducing the effective dataset size by a factor of approximately 2.8 (see Ablation Study for details).

### 3.3 TRAINING AND INFERENCE SETTINGS

For training, we used the BindingDB database from February 2025, containing 1,167,809 measurements after all cleaning and standardization steps (see Data Preparation for details). By optimizing the hyperparameter space, we found that the best quality is achieved with learning rate $5 \times 10^{-5}$, dropout 0.1, batch size 48, and the following decoder parameters: 8 layers, 8 heads and a hidden dimension 1,280. During the inference stage, we generated ligand sequences starting from fully masked sequences of a fixed length (170 tokens), sampling each masked token independently. To enable the model to adjust some tokens based on their contextual relationships, we used the re-masking technique introduced in Wang et al. (2025) and nucleus sampling introduced in Holtzman et al. (2019), both of which significantly reduced the number of invalid ligands generated. We evaluated all re-masking options described in Wang et al. (2025), and found that using nucleus sampling with a threshold of 0.9, the ReMDM-cap scheme with $\eta = 0.1$, and 250 sampling steps we achieve the best performance (see Ablation Study).

## 4 EXPERIMENTS

### 4.1 SETUP

To avoid data leakage and ensure a fair comparison with baseline models trained on different datasets, we did not use a conventional train-test split. Instead, following Liu et al. (2024), we selected 12 diverse protein targets for the test set, spanning the 7 most common protein families according to the ChEMBL protein classification [Davies et al. (2015)]. All target sequences from BindingDB were clustered using CD-HIT [Li & Godzik (2006)] at 60% identity.From the intersection of the CrossDocked2020 [Francoeur et al. (2020)] and BindingDB datasets, we chose 6 'easy' targets with over 1,000 training examples (ESR1, HCRTR1, JAK1, P2RX3, KDM1A, IDH1), and 6 'hard' targets with few examples (RIOK1, NR4A1, GRIK1, CCR9, FTO, SPIN1). See Table 4 for annotation details. For this table, we consider all protein-ligand pairs in a cluster as training examples for a target of that cluster.

To assess the overall performance of ProtoBind-Diff, we compared it with three recent generative models that sample molecules based on 3D protein pockets and have demonstrated strong performance: PocketFlow [Jiang et al. (2024)], Pocket2Mol [Peng et al. (2022)], and TargetDiff [Guan et al. (2023)]. We also added TamGen [Wu et al. (2024)] as a newer model that leverages pre-trained SMILES embeddings from PubChem [Kim et al. (2025)]. In addition, we selected REINVENT4 [Loeffler et al. (2024b)], a model that generates molecules based on desired chemical properties without conditioning on protein targets. For each target and model, we generated 1,000 SMILES strings to evaluate the percentage of unique, diverse, and novel molecules.

| | Validity (↑) | Uniqueness (↑) | Diversity (↑) | QED (↑) | SAScore (↓) | MMD (↓) |
|---|---|---|---|---|---|---|
| BindingDB (refer.) | $1.00 \pm 0.00$ | $1.00 \pm 0.00$ | $0.90 \pm 0.08$ | $0.55 \pm 0.07$ | $3.14 \pm 0.32$ | $0.00 \pm 0.00$ |
| ProtoBind-Diff | $0.72 \pm 0.11$ | $0.99 \pm 0.03$ | $\mathbf{1.00 \pm 0.01}$ | $0.58 \pm 0.06$ | $2.93 \pm 0.33$ | $\mathbf{0.11 \pm 0.11}$ |
| REINVENT4 | $0.85 \pm 0.10$ | $\mathbf{1.00 \pm 0.00}$ | $0.87 \pm 0.13$ | $\mathbf{0.64 \pm 0.11}$ | $\mathbf{2.32 \pm 0.26}$ | $0.31 \pm 0.17$ |
| Pocket2Mol | $0.81 \pm 0.24$ | $0.45 \pm 0.07$ | $0.79 \pm 0.10$ | $0.45 \pm 0.09$ | $3.94 \pm 0.65$ | $0.37 \pm 0.12$ |
| PocketFlow | $\mathbf{1.00 \pm 0.00}$ | $0.87 \pm 0.04$ | $0.99 \pm 0.01$ | $0.54 \pm 0.03$ | $2.88 \pm 0.27$ | $0.46 \pm 0.25$ |
| TamGen | $\mathbf{1.00 \pm 0.00}$ | $0.27 \pm 0.08$ | $0.87 \pm 0.04$ | $0.57 \pm 0.04$ | $3.06 \pm 0.42$ | $0.55 \pm 0.33$ |
| TargetDiff | $0.68 \pm 0.22$ | $\mathbf{1.00 \pm 0.00}$ | $\mathbf{1.00 \pm 0.00}$ | $0.34 \pm 0.12$ | $5.19 \pm 0.35$ | $0.69 \pm 0.25$ |

Table 1: Comparison of general chemical properties for generated molecules across all models. Each value is the average over 12 test targets. All properties, except validity, are computed after standardization and duplicate removal. Lower MMD values indicate greater similarity to the BindingDB reference set and thus better generation quality. Errors represent the values of standard error of the mean (SEM).

| | Fraction of Novel | Diversity (↑) | QED (↑) | SAScore (↓) | MMD (↓) |
|---|---|---|---|---|---|
| ProtoBind-Diff | $0.49 \pm 0.34$ | $\mathbf{1.00 \pm 0.01}$ | $0.61 \pm 0.05$ | $2.76 \pm 0.27$ | $\mathbf{0.18 \pm 0.10}$ |
| REINVENT4 | $0.84 \pm 0.24$ | $0.86 \pm 0.14$ | $\mathbf{0.65 \pm 0.11}$ | $\mathbf{2.29 \pm 0.24}$ | $0.33 \pm 0.17$ |
| Pocket2Mol | $0.21 \pm 0.11$ | $0.89 \pm 0.07$ | $0.54 \pm 0.09$ | $3.32 \pm 0.47$ | $0.37 \pm 0.21$ |
| PocketFlow | $0.82 \pm 0.05$ | $0.99 \pm 0.01$ | $0.54 \pm 0.03$ | $2.84 \pm 0.26$ | $0.47 \pm 0.26$ |
| TamGen | $0.25 \pm 0.09$ | $0.87 \pm 0.04$ | $0.57 \pm 0.04$ | $3.04 \pm 0.40$ | $0.57 \pm 0.33$ |
| TargetDiff | $0.75 \pm 0.15$ | $\mathbf{1.00 \pm 0.00}$ | $0.35 \pm 0.12$ | $5.06 \pm 0.34$ | $0.75 \pm 0.25$ |

Table 2: Comparison of general chemical properties for generated molecules after applying the novelty filter at threshold $T_{\text{sim}} = 0.5$. Values are shown for all generative models and reported as averages over 12 test targets. Lower MMD values indicate greater similarity to the BindingDB reference set and therefore better generation quality. The distribution of Tanimoto similarities $T_{\text{sim}}$ between generated molecules and BindingDB actives is shown in Figure 5.

## 4.2 PROPERTIES OF GENERATED MOLECULES

During the validation phase, some generated SMILES were found to be invalid or duplicated, as can be seen in Table 1. These samples were excluded from further analysis. Unlike other methods such as TamGen and Pocket2Mol, our model demonstrates reasonable diversity and uniqueness scores. For instance, TamGen produces more than $95\%$ valid molecules, but both uniqueness and diversity are relatively low. PocketFlow achieves the best performance in terms of validity and uniqueness, but it predominantly generates molecules with lower molecular weight compared to active compounds (see Table 1), suggesting a tendency to favor simpler structures. We also observed that validity can be improved by tuning the parameters of the re-masking sampler during the generation step; however, this comes at a trade-off against other molecular properties. We prioritize quality of generated molecules and diversity, since the number of valid molecules can be increased by running more inference batches.

One of the primary objectives in drug discovery is to generate novel compounds that are structurally distinct yet retain activity against a given protein target. To assess this, we evaluated model outputs under a novelty constraint, defining a molecule as novel if its maximum Tanimoto similarity ($T_{\text{sim}}$) to any active compound for the same target in BindingDB is less than 0.5. Tanimoto similarity quantifies the overlap between binary molecular fingerprints: a value of 1 denotes identical compounds, whereas 0 denotes no shared features. We report the fraction of such structurally novel molecules as the Fraction of Novel value, and evaluate how specific these molecules are to the target (Table 2).

We observe that ProtoBind-Diff and Pocket2Mol tend to generate molecules highly similar to known actives on 'easy' targets, which is reflected in Tanimoto similarity histograms skewed toward 1 (see Figure 5). Conversely, for 'hard' targets, these models generate compounds with lower similarity, shifting histograms toward 0. We interpret high similarity as a potential sign of overfitting. However, complete dissimilarity may suggest a lack of protein-specific conditioning, an issue particularly evident in models such as PocketFlow and TamGen (Table 2). Furthermore, unconditional

|  | Vina (docking score) | Boltz-1 (ipTM score) | Boltz-2 (binary probability) |
|---|---|---|---|
| BindingDB (active) | $3.21 \pm 1.40$ | $6.28 \pm 1.75$ | $7.78 \pm 1.72$ |
| ProtoBind-Diff | $1.21 \pm 0.45$ | $\mathbf{2.30 \pm 0.47}$ | $\mathbf{3.40 \pm 0.91}$ |
| REINVENT4 | $1.44 \pm 0.51$ | $1.06 \pm 0.25$ | $0.52 \pm 0.15$ |
| Pocket2Mol | $\mathbf{5.50 \pm 2.20}$ | $2.26 \pm 0.42$ | $3.07 \pm 0.93$ |
| PocketFlow | $2.40 \pm 0.74$ | $1.37 \pm 0.26$ | $1.20 \pm 0.42$ |
| TamGen | $0.50 \pm 0.25$ | $1.89 \pm 0.50$ | $0.82 \pm 0.22$ |
| TargetDiff | $0.49 \pm 0.30$ | $1.50 \pm 0.22$ | $1.36 \pm 0.37$ |

Table 3: Enrichment Factor (EF) analysis of AutoDock Vina (the first column), Boltz-1 (the second column) and Boltz-2 (the third column) scorers for identifying active molecules above thresholds compared to randomly selected active molecules from BindingDB. Thresholds used: AutoDock Vina docking score $< -10$ kcal/mol, Boltz-1 ipTM score $> 0.85$ and Boltz-2 affinity probability binary value $> 0.5$. Errors represent the values of standard error of the mean (SEM). Data per target is presented in Tables 7-9.

REINVENT4 model faces challenges in achieving an optimal similarity balance. REINVENT4 also tends to generate molecules with higher drug-likeness (QED) and lower synthesizability (SAScore) than reference actives, indicating a preference for chemically simpler compounds (Table 1 and Figure 6). This may reflect a bias toward general drug-likeness, placing model's outputs further from the distribution of known actives.

The performance of a conditional generative model is best assessed by how accurately its generated molecular distribution recapitulates a ground-truth distribution. To this end, we computed the Maximum Mean Discrepancy (MMD) across a set of key molecular properties (detailed in Section Chemical properties). MMD quantifies the divergence between two distributions, where a lower value signifies a closer match to the properties of real molecules. ProtoBind-Diff consistently outperforms nearly all competing models across the individual descriptors (Table 6). The sole exception is REINVENT4, a model that is not target-specific and is explicitly designed to optimize these properties by construction. Consequently, our model yields an overall distribution of molecular properties that more closely mirrors that of the reference compounds, even for novel molecules. Mean molecular property values for each target are shown in Figure 6.

### 4.3 Structure-Based Evaluation of Generated Ligands

In the absence of experimental binding affinity data for the generated molecules, we evaluated their structural plausibility using two distinct approaches: classical molecular docking with AutoDock Vina and models Boltz-1/Boltz-2 for biomolecular interaction prediction.

Docking was performed using the standard AutoDock Vina protocol, with the binding site defined by the position of reference ligands in experimentally determined structures from the Protein Data Bank (PDB) [Berman et al. (2000)]. All generative models performed well on targets where docking effectively distinguished active from inactive compounds, for example, ESR1, GRIK1, and CCR9 (Figure 7). However, in most cases, docking exhibited poor discriminatory power. For example, with targets such as P2RX3, KDM1A, IDH1, RIOK1, NR4A1, FTO and SPIN1, the difference in the average docking scores between active and inactive molecules was not statistically different. Notably, for several of these targets, Pocket2Mol and REINVENT4 achieved significantly lower docking scores than all other models and even true active compounds, e.g., KDM1A, IDH1, RIOK1 and SPIN1.

For all methods, the docking scores of generated compounds varied substantially across targets. Overall docking performance is summarized in Table 3 using the enrichment factor (EF), which quantifies whether the concentration of predicted active molecules in the observed set is higher (EF $> 1$) or lower (EF $< 1$) than in the reference set. EF was computed as EF$= C_a^{\text{gen}}/C_a^{\text{rand}}$, where $C_a^{\text{gen}}$ and $C_a^{\text{rand}}$ denote the fractions of active molecules (defined as having a Vina score $< -10$ kcal/mol) in the generated set and in the random subset of all active molecules from BindingdDB, respectively.

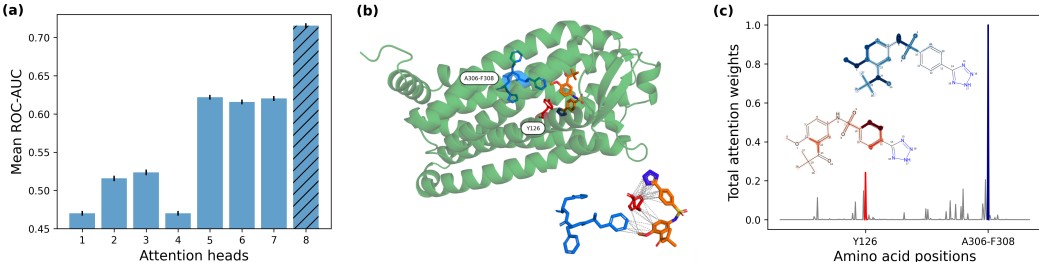

Figure 2: Interpretability of ProtoBind-Diff attention. **(a)** Mean ROC-AUC ($\pm$ SEM) for binding-site detection for the eight attention heads of ProtoBind-Diff, averaged over 1,843 annotated sequences. Head 8 shows the highest ROC-AUC of 0.72. **(b)** Predicted pose of a ligand generated by ProtoBind-Diff (orange) in complex with CCR9 protein (green). The structure was generated using Boltz-1 model. Inset: Predicted binding interactions between the ligand and amino acid contact residues, based on a 5 Å distance cutoff. **(c)** Attention weights from head 8 of ProtoBind-Diff, averaged over ligand tokens and plotted against residue positions in the protein sequence. Atoms in the molecular graph are colored with intensity proportional to their attention weights. Peaks in the amino acid sequence align with residues that are in direct contact with the ligand in the predicted pose, suggesting that the model's attention mechanism captures spatially relevant interaction signals from sequence alone.

Based on docking EFs, ProtoBind-Diff ranked below Pocket2Mol and PocketFlow. Notably, Pocket2Mol exhibited a surprisingly high EF, surpassing even that of the true active molecules. We attribute this to the fact that both Pocket2Mol and PocketFlow were trained on the CrossDocked2020 dataset, which, although based on crystallographic structures, was heavily augmented (by a factor of 100) with Vina-generated poses. This likely led to overfitting, causing the models to preferentially generate molecules that score well under Vina. Conversely, the relatively low EF observed for true actives suggests that the Vina scoring function may not align well with actual binding activity.

To complement docking, we applied the Boltz-1 and Boltz-2, recent open source deep learning models for protein-ligand structure prediction, to the same sets of generated molecules and targets. Boltz-1 was used to predict ligand-protein complexes, providing an interface predicted TM-score (ipTM), a confidence metric that estimates the structural plausibility of the predicted binding interface. Boltz-2 is the next-generation model in the Boltz family, offering improved structural accuracy, significantly faster performance, and the added capability of binding affinity prediction. Across targets, the boxplots (Figures 8 and 9) show Boltz-2's affinity probability yields the strongest separation between actives and inactives, Boltz-1's ipTM is second, and both exceed docking. Because both ProtoBind-Diff and Boltz-2 affinity model were trained on BindingDB, we treat Boltz-2 as potentially biased and use Boltz-1's ipTM as the primary metric.

For nearly all 'easy' targets (ESR1, HCRTR1, JAK1, KDM1A, IDH1, P2RX3), ProtoBind-Diff produced ipTM score distributions that were comparable to or better than those of structure-based models, including PocketFlow, Pocket2Mol, and TargetDiff. On 'hard' targets, ProtoBind-Diff achieved the top or near-top ipTM scores for SPIN1, GRIK1, RIOK1, CCR9 and NR4A1. Enrichment results using Boltz-1 (Table 3) further show stronger discrimination of actives than docking, with ProtoBind-Diff achieving the highest EF, closely followed by Pocket2Mol.

## 4.4 ATTENTION-BASED BINDING SITE ANALYSIS

To investigate whether ProtoBind-Diff captures interpretable patterns of protein-ligand interaction, we analyzed the attention heads in the final decoder layer using 1,843 BioLiP-2 annotated proteins [Zhang et al. (2024a)]. Attention weights were used as unsupervised predictors of binding site residues (see Attention Visualization and Docking Analysis). Figure 2**a** presents a bar plot showing the mean ROC-AUC for each attention head, with the standard errors of the mean (SEM) across the annotated proteins. Attention head 8 yielded the highest ROC-AUC of $0.716 \pm 0.003$. For comparison, a linear classifier trained in a supervised manner on ESM-2 embeddings (using a similarity-based train/test split) yielded a ROC-AUC of $0.849 \pm 0.003$. Since ProtoBind-Diff was not

trained on BioLiP-2 residue labels or other binding pocket data, the strong performance of attention head 8 indicates that the model independently learns to focus on structurally relevant regions without explicit supervision. This suggests that the model's attention mechanism encodes biophysically meaningful patterns.

Figure 2 illustrates a representative case study involving GPCR CCR9, with a ligand generated by ProtoBind-Diff that achieved a high Boltz-1 ipTM score. The protein-ligand complex was predicted using the Boltz-1 model based on the CCR9 receptor sequence from PDB entry 5LWE. The predicted binding pose highlights specific contact residues surrounding the ligand (Figure 2**b**). We further extracted attention maps from attention head 8 of the final transformer block, averaging the weights across all ligand tokens to derive a per-residue weight vector. Remarkably, the attention profile over the protein sequence (Figure 2**c**) exhibits distinct peaks at amino acid positions that are close to the contact residues (Figure 2**b**) and also captures ligand substructures aligned with these residues. We further quantified the contributions of individual atoms in the ligand based on their attention weight values. These results demonstrate that ProtoBind-Diff's cross-attention mechanism effectively integrates protein sequence information and ligand structural features, aligning with biophysically meaningful interactions that mediate ligand recognition.

## 5 Conclusion and Future Works

We proposed ProtoBind-Diff, a discrete diffusion model designed for ligand generation conditioned on protein sequences via pre-trained ESM-2 embeddings. Trained on over one million BindingDB pairs, the model generates valid, drug-like molecules whose physicochemical profiles closely match those of known actives, while maintaining high novelty and scaffold diversity. On a 12-target benchmark spanning frequently and sparsely represented proteins, ProtoBind-Diff attains strong affinity metrics on Boltz-1 scores and outperforms baselines trained on 3D structures. Furthermore, we demonstrate that model attention weights align with binding site residues, suggesting genuine exploitation of sequence context rather than memorization. This represents a paradigm shift from previous target-aware molecular generation approaches, which rely on 3D structures and pocket selection. Moreover, baselines trained on 3D data often yield docking scores better than those of true actives, indicating optimization toward the docking objective rather than binding affinity. These results demonstrate that sequence-only conditioning is a viable and scalable method for ligand design across the proteome, including targets that lack reliable structures. Looking ahead, we plan to (i) improve the quality of generated molecules by integrating other available ligand–protein datasets containing bioactivity information, such as Papyrus [Béquignon et al. (2023)] (ii) extend conditioning to protein families or complexes, (iii) try more robust and interpretable molecular representations, such as SELFIES or SAFE, (iv) include recent approaches such as FlexSBDD and DynamicFlow to the benchmark, and (v) combine ProtoBind-Diff with reinforcement learning or preference optimization using structure-based evaluators such as Boltz-2 as rewards or preference models, in the spirit of recent work on RL for SDEs and direct preference optimization in SBDD [Zhou et al. (2024a;b); Cheng et al. (2024)].

## Data Availability

All data used in this study are publicly available. Molecule-protein interaction data were obtained from the BindingDB database (`https://www.bindingdb.org`), an open-access resource for binding affinity data. No proprietary datasets were used.

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

| Target | Samples (train set) | Dataset type | Protein class | UniProt ID | PDB | L1 family | L2 family |
|---|---|---|---|---|---|---|---|
| ESR1 | 4483 | easy | Nuclear receptor | P03372 | 2r6w | Transcription factor | Nuclear receptor |
| HCRTR1 | 12691 | easy | GPCR | O43613 | 4zjc | Membrane receptor | Family A GPCR |
| JAK1 | 12455 | easy | Kinase | P23458 | 3eyg | Enzyme | Kinase |
| P2RX3 | 5140 | easy | Ion channel | P56373 | 5svl | Ion channel | Ligand-gated ion channel |
| KDM1A | 4622 | easy | Protein-protein interaction target | O60341 | 5lhg | Epigenetic regulator | Eraser |
| IDH1 | 5177 | easy | Non-kinase enzyme | O75874 | 4umx | Enzyme | Oxidoreductase |
| RIOK1 | 15 | hard | Kinase | Q9BRS2 | 4otp | Enzyme | Kinase |
| NR4A1 | 28 | hard | Nuclear receptor | P22736 | 3v3q | Transcription factor | Nuclear receptor |
| GRIK1 | 335 | hard | Ion channel | P39086 | 3fv1 | Ion channel | Ligand-gated ion channel |
| CCR9 | 82 | hard | GPCR | P51686 | 5lwe | Membrane receptor | Family A GPCR |
| FTO | 37 | hard | Non-kinase enzyme | Q9C0B1 | 4zs3 | Enzyme | Oxidoreductase |
| SPIN1 | 19 | hard | Protein-protein interaction target | Q9Y657 | 5jsj | Epigenetic regulator | Reader |

Table 4: Table with annotation of chosen proteins for the test set. For each target, we listed the number of training samples available, the type of dataset (easy/hard), the protein family name, the UniProt identifier, the PDB code, and the first- and second-level family names (L1 and L2) according to the ChEMBL classification.

## A  DATA PREPARATION

We used the BindingDB database from February 2025, containing 3,010,313 measurements, of which 1,311,211 unique compounds, 9,524 unique targets from various assays and families. Ligands were presented in SMILES format and proteins were presented in the form of amino acid sequences. Before training, the data was cleaned using the following procedure:

1. Sequences lacking a UniProt ID (according to the EMBL-EBI database), with unknown organism source, or belonging to very rare clusters were removed,

2. Cytochrome P450 and Albumin were excluded from the analysis due to their non-specific binding to all ligands,

3. All invalid SMILES and SMILES containing very rare tokens (occurring fewer than 100 times in the original dataset) were removed.,

4. Only sequences with lengths between 50 and 1,500 amino acids and ligands containing between 10 and 80 atoms were retained. Very short sequences do not form stable binding pockets, while very long protein chains and large ligands were excluded to ensure the model fit into GPU memory.

For the purpose of replicating the true data distribution during training, only active instances were chosen. Binary labels were subsequently assigned based on the following criterion: a molecule was classified as active if at least one of the $K_i$, $K_d$ or $EC_{50}$ values exhibited an activity below 1 $\mu$M. The resulting dataset included 1,167,809 samples.

Protein sequences were encoded using a pre-trained protein language model. ESM-2 embeddings, characterized by 650 million parameters, 33 model layers, and a dimension of 1,280, were selected for this purpose. The SMILES-formatted ligand sequences were converted to tokens using the PyS-MILESUtils library [Bjerrum et al. (2021)].

| Model | ESM-2 model | $T_{sim}$ | Validity | QED | SAScore | MMD |
|---|---|---|---|---|---|---|
| No rand. + no sampl. | 650M | 1.0 | 0.81±0.04 | 0.49±0.05 | 2.68±0.10 | 0.23±0.06 |
| No rand. + sampl. | 650M | 1.0 | 0.51±0.01 | 0.48±0.00 | 2.93±0.01 | 0.35±0.06 |
| Rand. + sampl. | 650M | 1.0 | 0.70±0.04 | 0.49±0.05 | 2.84±0.08 | 0.19±0.05 |
| Rand. + sampl. | 150M | 1.0 | 0.87±0.00 | 0.62±0.00 | 2.52±0.01 | 0.48±0.08 |

| Model | ESM-2 model | $T_{sim}$ | Fraction of Novel | QED | SAScore | MMD |
|---|---|---|---|---|---|---|
| No rand. + no sampl. | 650M | 0.5 | 0.45±0.11 | 0.55±0.04 | 2.50±0.07 | 0.40±0.10 |
| No rand. + sampl. | 650M | 0.5 | 0.78±0.04 | 0.50±0.00 | 2.85±0.02 | 0.45±0.10 |
| Rand. + sampl. | 650M | 0.5 | 0.50±0.09 | 0.51±0.05 | 2.76±0.07 | 0.28±0.08 |
| Rand. + sampl. | 150M | 0.5 | 0.87±0.03 | 0.63±0.00 | 2.45±0.02 | 0.55±0.11 |

Table 5: Chemical properties of generated molecules from models with different parameters and protein embeddings. Properties are shown for all generated molecules (the first block) as well as for those that passed through the novelty filter with Tanimoto threshold $T_{sim} = 0.5$ (the second block). All values were calculated separately for all targets from the separate test set (CA12, DHODH, GLS, BRD4, TEK, GCKR, PRSS2, TACR3) and then averaged. Errors represent the values of standard error of the mean (SEM).

# B   ABLATION STUDY

To find optimal parameter values, the models were tested on a subset of targets not included in our benchmark, and consisted of both common and rare proteins from BindingDB with the following L2 families annotation: CA12 (Lyase), DHODH (Oxidoreductase), GLS (Hydrolase), BRD4 (Reader), TEK (Kinase), GCKR (Enzyme), PRSS2 (Protease), and TACR3 (Family A G protein-coupled receptor). A different test set was chosen so as not to interfere with the results of comparison of our model with baselines. For each model 1,000 molecules were generated for each target and the results were then averaged between them.

**Model parameters.** We compared model configurations by their ability to generate novel, unique molecules whose properties resemble those of known actives for the same protein. To isolate the effect of SMILES randomization and molecular sampling strategies, we trained models with and without these modifications. After filtering for valid molecules, we computed the Tanimoto similarity between each generated compound and all known actives for the corresponding target. Metrics were reported both over all valid molecules and over the subset with maximum Tanimoto similarity less than 0.5. As shown in Table 5, all models, except the one trained with sampling only, exhibit no

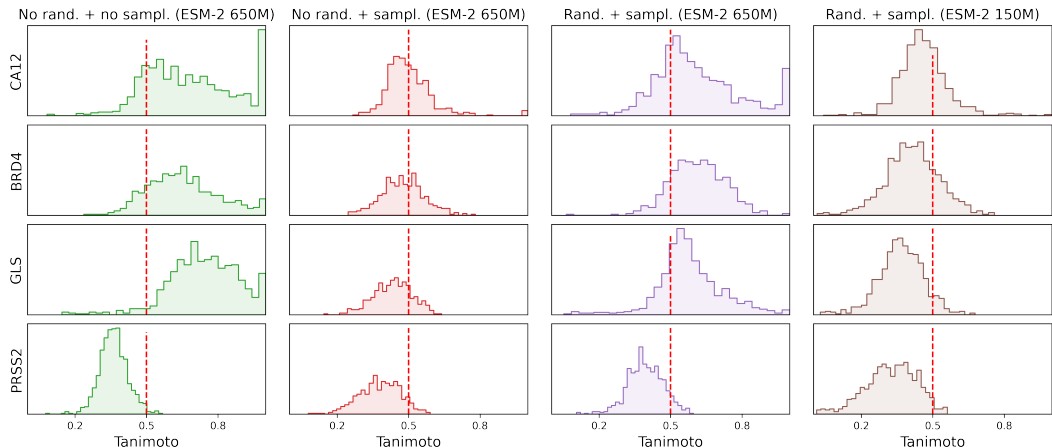

Figure 3: Maximum Tanimoto similarity between generated molecules and BindingDB actives for targets CA12, BRD4, GLS, and PRSS2, across different model and protein embedding configurations. The red dashed line denotes the novelty threshold $T_{\mathrm{sim}} = 0.5$.

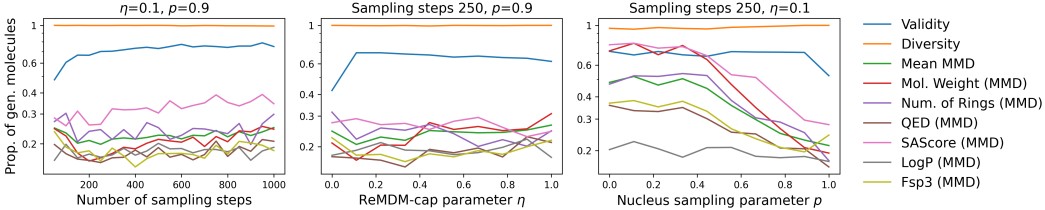

Figure 4: Properties of generated molecules obtained with different parameters of the sampler. All values were calculated separately for all targets (CA12, DHODH, GLS, BRD4, TEK, GCKR, PRSS2, TACR3) and then averaged.

significant differences in QED, SAScore, or MMD (computed from multiple chemical descriptors; see Section Chemical properties). However, without sampling the model strongly overfits to training set scaffolds and fails to produce novel molecules (Figure 3). We also evaluated two pre-trained ESM-2 embeddings: one with 650M parameters (33 layers and dimention 1280) and another with 150M parameters (30 layers and dimention 640). The smaller embedding tended to yield more valid molecules but with higher MMD (Table 5). To strike a balance between quality and computational efficiency, we chose embeddings with an intermediate size having 650 million parameters. Finally, uniqueness and diversity did not differ significantly across models.

**Sampler parameters.** For this experiment, we chose a model with an ESM-2 embedding having 650 million parameters, enabled randomization, and clustering of molecules. To generate samples for all targets, two sampler parameters ($\eta$, ReMDM-cap scheme; $p$, nucleus parameter; or number of steps) were fixed, while the third variable was varied. Subsequently, validity, uniqueness, and MMD metric were calculated for each generated sample. Notably, only the validity metric is affected by the increase in the number of sampling steps (Figure 4). As the time required for generation increases proportionally with the number of sampling steps, we selected an optimal value of 250 to facilitate rapid generation without compromising validity. In Figure 4, it can be seen that in order to generate molecules with the desired properties, it is better to choose lower values of $\eta$ and higher values of nucleus sampling parameter $p$.

## C    CHEMICAL PROPERTIES

To estimate the quality of generated molecules, we computed the following metrics: (1) Validity, which is the proportion of valid molecules among all generated candidates; (2) Uniqueness, which

is the fraction of unique SMILES strings in their canonical form; (3) FracNovel, which is the the fraction of molecules with Tanimoto similarity less than 0.5 to the reference molecules; (4) Diversity, which is the number of unique clusters using Taylor-Butina [Taylor (1995)] clustering algorithm with Tanimoto similarity cutoff 0.2 divided by the total number of samples.

In addition, we computed the following molecular descriptors: (5) Molecular weight; (6) LogP (octanol-water partition coefficient); (7) Number of rotatable bonds; (8) TPSA (topological polar surface area); (9) Number of rings; (10) QED (quantitative estimate of drug-likeness) [Bickerton et al. (2012)]; (11) SAScore (synthetic accessibility score) [Ertl & Schuffenhauer (2009)]; (12) Number of heavy (non-hydrogen) atoms; (13) Number of aromatic rings; (14) CSP3 (fraction of sp3-hybridized carbons) [Lovering et al. (2009)]. Descriptors (5)-(14) were used to compute the Maximum Mean Discrepancy (MMD) between the generated and reference sets. The closer these distributions are to those of known actives, the better the generation quality. All metrics and descriptors, except validity, were computed after standardization and duplicate removal, using the open source cheminformatics library RDKit (https://www.rdkit.org).

| | $T_{sim}$ | Weight | LogP | Rot. Bonds | TPSA | Rings | QED | SA Score | Heavy Atoms | Arom. Rings | Fsp3 | Avg. |
|---|---|---|---|---|---|---|---|---|---|---|---|---|
| ProtoBind-Diff | 1.0 | **0.09** | **0.08** | **0.09** | **0.13** | **0.12** | **0.11** | **0.11** | **0.10** | **0.18** | **0.11** | **0.11** |
| REINVENT4 | 1.0 | 0.15 | **0.08** | 0.30 | 0.24 | 0.64 | 0.20 | 0.71 | 0.17 | 0.26 | 0.30 | 0.31 |
| Pocket2Mol | 1.0 | 0.37 | 0.25 | 0.39 | 0.27 | 0.33 | 0.34 | 0.51 | 0.39 | 0.54 | 0.36 | 0.37 |
| PocketFlow | 1.0 | 0.48 | 0.22 | 0.27 | 0.78 | 0.75 | 0.16 | 0.28 | 0.49 | 0.70 | 0.40 | 0.46 |
| TamGen | 1.0 | 0.73 | 0.52 | 0.29 | 0.20 | 1.03 | 0.15 | 0.29 | 0.81 | 1.22 | 0.26 | 0.55 |
| TargetDiff | 1.0 | 0.29 | 0.29 | 0.56 | 0.39 | 0.55 | 0.61 | 1.32 | 0.29 | 1.88 | 0.77 | 0.69 |
| | $T_{sim}$ | Weight | LogP | Rot. Bonds | TPSA | Rings | QED | SA Score | Heavy Atoms | Arom. Rings | Fsp3 | Avg. |
| ProtoBind-Diff | 0.5 | 0.18 | 0.10 | **0.12** | **0.21** | **0.28** | 0.17 | **0.19** | 0.20 | **0.25** | **0.14** | **0.18** |
| REINVENT4 | 0.5 | **0.16** | **0.09** | 0.32 | 0.25 | 0.69 | 0.21 | 0.79 | **0.17** | 0.29 | 0.32 | 0.33 |
| Pocket2Mol | 0.5 | 0.45 | 0.25 | 0.38 | 0.29 | 0.34 | 0.26 | 0.32 | 0.42 | 0.63 | 0.38 | 0.37 |
| PocketFlow | 0.5 | 0.52 | 0.22 | 0.28 | 0.82 | 0.81 | 0.16 | 0.28 | 0.53 | 0.72 | 0.40 | 0.47 |
| TamGen | 0.5 | 0.76 | 0.53 | 0.29 | **0.21** | 1.11 | **0.14** | 0.29 | 0.84 | 1.28 | 0.27 | 0.57 |
| TargetDiff | 0.5 | 0.31 | 0.30 | 0.61 | 0.39 | 0.80 | 0.60 | 1.26 | 0.32 | 2.07 | 0.85 | 0.75 |

Table 6: Values of Maximum Mean Discrepancy (MMD) metric from generated molecules to reference molecules from BindingDB. All metrics are calculated for both all generated molecules (the first block) and those that passed through the novelty filter (the second block). The results are displayed for all generative models, with each number representing the average performance over 12 test targets. Lower MMD value means greater similarity to the reference dataset BindingDB and indicates better generative quality. The Tanimoto similarity ($T_{sim}$) between molecules generated by all models and active molecules from BindingDB can be seen in Figure 5.

## D  DOCKING EVALUATION

For each benchmark target and generative model, we generated 1,000 molecules and applied a consistent selection protocol to obtain the final set. After filtering for valid molecules, we computed the Tanimoto similarity between each generated compound and all known actives for the same target. Only molecules with a maximum similarity below 0.5 were retained, ensuring structural novelty with respect to known ligands. From this subset, we randomly sampled up to 100 unique molecules per target. We used DockStream [Guo et al. (2021)], a molecular docking wrapper for Python, for the automated preparation of targets, ligand embedding, and docking. The processed crystal structures and the reference ligand for pocket detection were taken from CrossDocked2020 [Francoeur et al. (2020)] dataset. The PDB accession codes for each test target are presented in Table 4. The grid box size was 20 X 20 X 20 Å centered on the position of the center of mass of the reference ligand. The docking scores of the generated molecules were obtained for each target using AutoDock Vina version 1.2.5 [Eberhardt et al. (2021)] with default parameters unless otherwise specified.

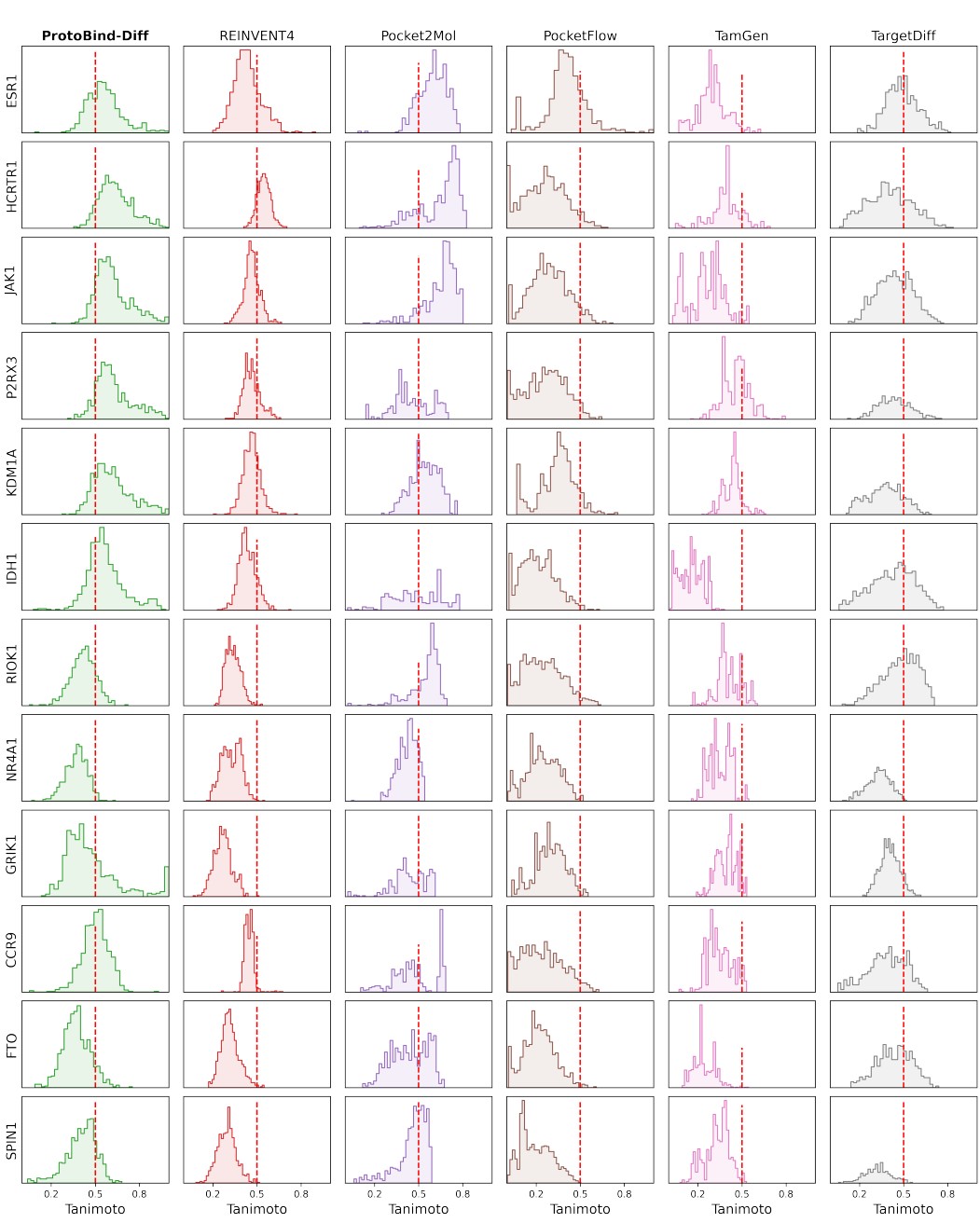

Figure 5: Maximum Tanimoto similarity between molecules generated by different models across 12 benchmark protein targets and known active molecules from BindingDB. Red dashed line indicates the novelty threshold ($T_{\mathrm{sim}} = 0.5$).

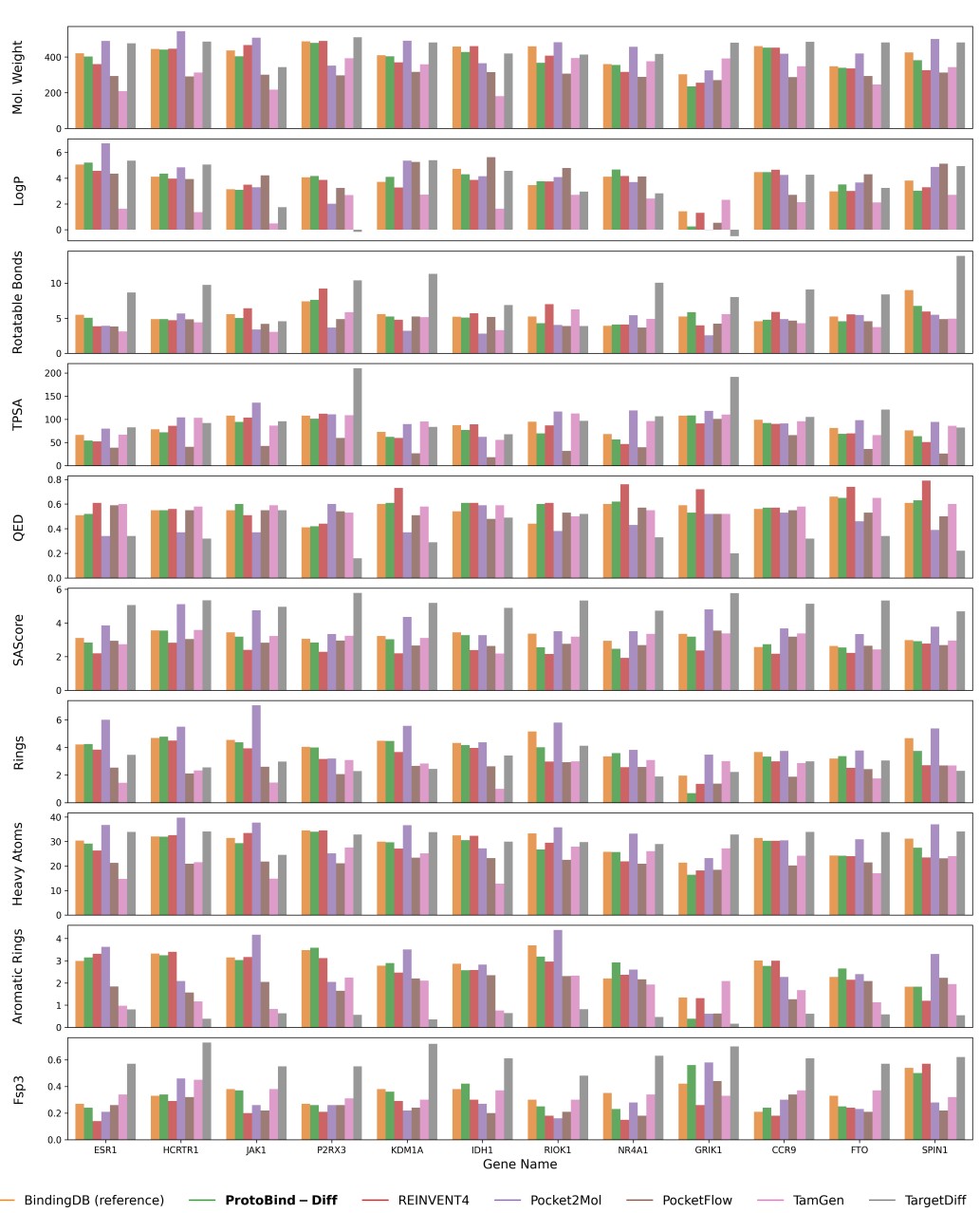

Figure 6: The main chemical properties of generated molecules for all targets separately from the test dataset grouped by generative models.

## E  BOLTZ EVALUATION

Boltz scores were calculated using the publicly available Boltz-1 and Boltz-2 pre-trained models, which integrate ligand preparation, pose generation, and scoring into a reproducible workflow. Boltz-1 was trained on all PDB protein-ligand complexes released before September 30, 2021, with a resolution of at least 9 Å, as described by Wohlwend et al. (2024). Boltz-2 was trained on PDB structures with a cutoff of June 1, 2023, as well as artificial samples and binding affinity data (in particular, PubChem, ChEMBL, and BindingDB) [Passaro et al. (2025)].

To make predictions with both models, we used PDB protein sequences as inputs. In the case of Boltz-1 model, each ligand was scored based on the interface TM-score (ipTM) for ligand-protein complex, which estimates the confidence of interfacial structural alignment. We observed that active molecules consistently show higher ipTM scores than inactive ones (see Figure 8), indicating that the Boltz-1 interface score can discriminate binders from non-binders. For Boltz-2, we used the binary affinity probability, which distinguishes actives from inactives well (see Figure 9). Statistical comparisons and molecules preprocessing were carried out in an analogous manner to that employed for docking protocols.

## F  ATTENTION VISUALIZATION AND DOCKING ANALYSIS

From the set of canonical protein sequences of the training set, we selected those that have binding site annotations in BioLiP-2 [Zhang et al. (2024a)], which resulted in 1843 sequences. For each selected sequence, we passed an active molecule through the model and extracted the attention weights of the final layer of the decoder. For every generated ligand, attention scores were averaged over ligand tokens to obtain a per-residue weight vector. Canonical sequences were segmented into non-overlapping 3-residue windows; each window was assigned the maximum weight among its residues. This step allows us to treat near-misses as successful binding site detections. A window was labeled 'positive' if any of its residues overlapped a BioLiP-annotated binding site. We then computed per-protein ROC-AUC from the window-level labels and scores. The resulting mean ROC-AUCs across all selected proteins are reported in Figure 2**a**.

To compare our results to a baseline, we trained a simple logistic regression on ESM-2 embeddings, using a train-test split based on protein sequence dissimilarity. Protein sequences were clustered using CD-HIT [Li & Godzik (2006)] at 60% identity, and clusters were randomly divided into train and test sets. After training the model to predict binding site residues, we evaluated its performance on the test set using ROC-AUC, following the same windowing procedure. To interpret the attention weights of our model, we sampled molecules for the C-C chemokine receptor type 9 (CCR9) protein. Subsequently, a representative molecule with a high Boltz-1 ipTM score for ligand-protein complex was selected. We focused on attention head 8 since it showed the best ROC-AUC values across all the analyzed heads. We identified peaks in the resulting profile and highlighted the corresponding values on the molecule using RDKit; these were then compared to the residues in contact with the ligand's docked pose, which was visualized using PyMOL [Schrödinger & DeLano].

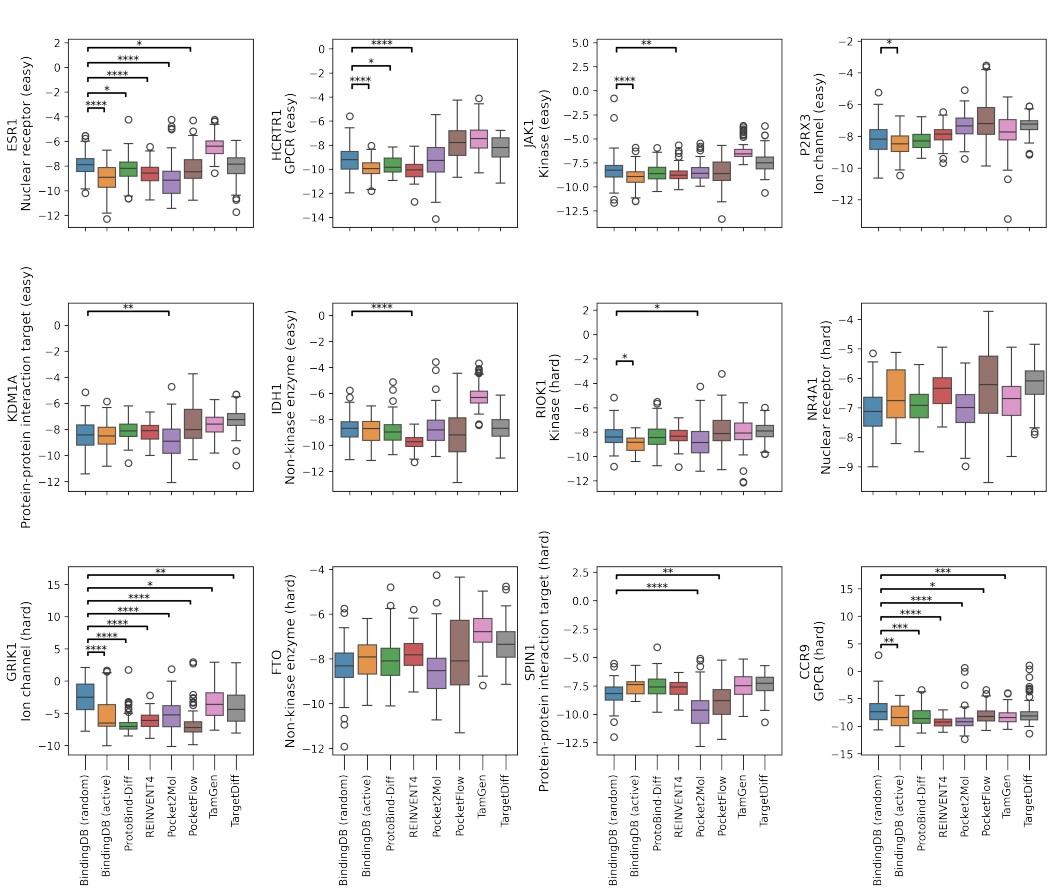

Figure 7: Docking scores of molecules generated by different models for 12 benchmark targets. Each boxplot shows the distribution of docking scores (lower is better). Statistical differences between selected model pairs were tested using the two-sided Mann-Whitney U test. Significance thresholds for adjusted p-values (Bonferroni correction): $p < 0.05$ (*), $< 0.01$ (**), $< 0.001$ (***), $< 0.0001$ (****).

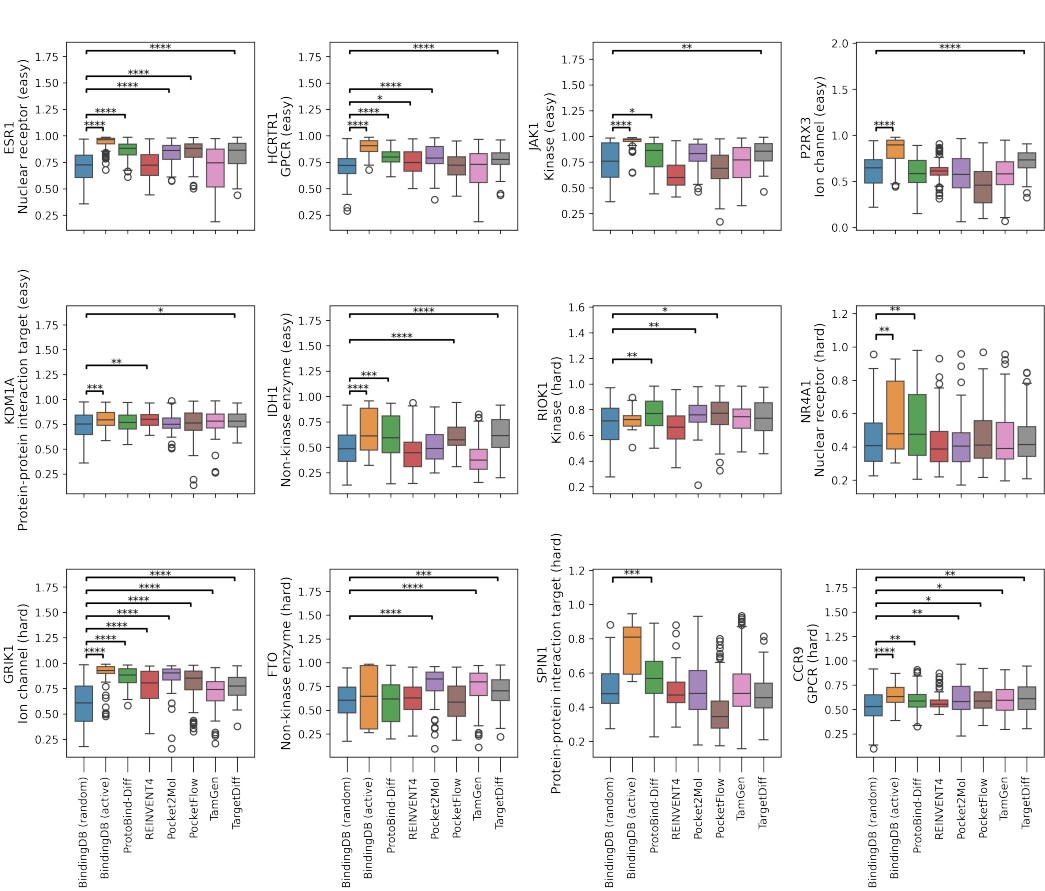

Figure 8: Boltz-1 scores of molecules generated by different models across 12 benchmark protein targets. Each boxplot shows the distribution of Boltz-1 scores for generated ligands targeting a specific protein, grouped by generative model (higher is better). Targets are categorized as 'easy' (top two rows) or 'hard' (bottom two rows) based on training set coverage. Statistical comparisons were performed using two-sided Mann-Whitney U tests. Significance thresholds for adjusted p-values (Bonferroni correction): $p < 0.05$ (*), $< 0.01$ (**), $< 0.001$ (***), $< 0.0001$ (****).

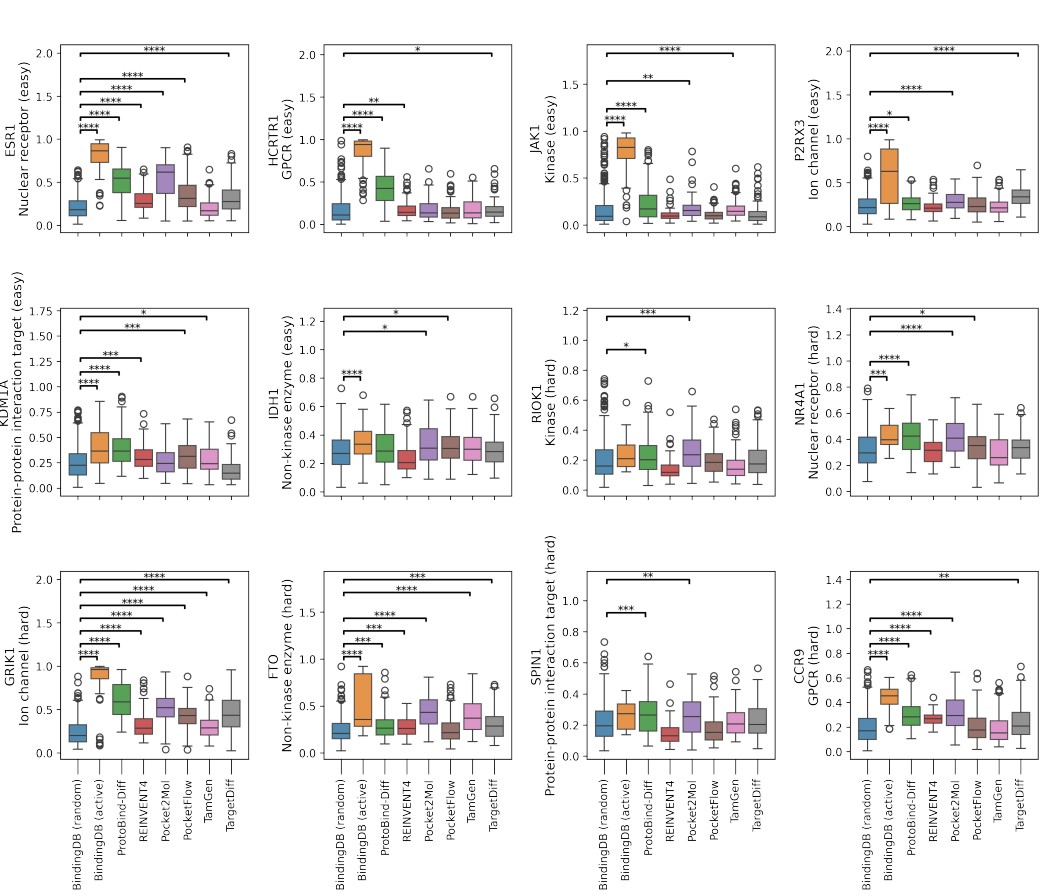

Figure 9: Boltz-2 binary affinity probability scores of molecules generated by different models across 12 benchmark protein targets. Each boxplot shows the distribution of Boltz-2 scores for generated ligands targeting a specific protein, grouped by generative model (higher is better). Targets are categorized as 'easy' (top two rows) or 'hard' (bottom two rows) based on training set coverage. Statistical comparisons were performed using two-sided Mann-Whitney U tests. Significance thresholds for adjusted p-values (Bonferroni correction): $p < 0.05$ (*), $< 0.01$ (**), $< 0.001$ (***), $< 0.0001$ (****).

| | ESR1* | HCRTR1* | JAK1* | P2RX3 | KDM1A | IDH1 | RIOK1 | NR4A1 | GRIK1* | FTO | SPIN1 | CCR9* |
|---|---|---|---|---|---|---|---|---|---|---|---|---|
| BindingDB (active) | 18.4 | 1.7 | 2.1 | 0.5 | 0.6 | 1.1 | 6.9 | 0.0 | 1.0 | 0.5 | 0.0 | 5.5 |
| ProtoBind-Diff | 4.9 | 1.2 | 0.4 | 0.0 | 0.1 | 0.9 | 2.9 | 0.0 | 0.0 | 0.2 | 0.0 | 3.8 |
| REINVENT4 | 4.9 | 2.1 | 1.0 | 0.0 | 0.1 | 3.1 | 1.0 | 0.0 | 0.0 | 0.0 | 0.0 | 5.2 |
| Pocket2Mol | 25.4 | 1.2 | 0.0 | 0.0 | 3.3 | 1.4 | 19.2 | 0.0 | 1.0 | 1.3 | 8.2 | 4.9 |
| PocketFlow | 7.8 | 0.5 | 2.8 | 0.0 | 0.3 | 3.3 | 7.0 | 0.0 | 0.0 | 1.1 | 4.9 | 1.2 |
| TamGen | 0.0 | 0.0 | 0.0 | 0.7 | 0.0 | 0.0 | 2.9 | 0.0 | 0.0 | 0.0 | 0.4 | 1.9 |
| TargetDiff | 4.0 | 0.3 | 0.2 | 0.0 | 0.1 | 0.7 | 0.0 | 0.0 | 0.0 | 0.0 | 0.2 | 0.5 |

Table 7: Enrichment factors for each target based on results of docking. *Indicates targets with a significant difference ($p < 0.05$) in Vina docking scores between active and random subsets of BindingDB (see Figure 7).

| | ESR1* | HCRTR1* | JAK1* | P2RX3* | KDM1A* | IDH1* | RIOK1 | NR4A1 | GRIK1* | FTO | SPIN1 | CCR9* |
|---|---|---|---|---|---|---|---|---|---|---|---|---|
| BindingDB (active) | 4.0 | 7.6 | 2.4 | 8.0 | 1.5 | 4.7 | 1.2 | 10.0 | 6.4 | 4.1 | 25.0 | 0.6 |
| ProtoBind-Diff | 2.8 | 2.8 | 1.3 | 1.0 | 1.0 | 3.2 | 1.6 | 6.6 | 4.4 | 1.5 | 1.0 | 0.5 |
| REINVENT4 | 0.8 | 2.5 | 0.2 | 0.6 | 1.1 | 0.7 | 0.5 | 0.5 | 3.2 | 1.4 | 1.0 | 0.2 |
| Pocket2Mol | 2.4 | 3.2 | 1.1 | 1.8 | 0.7 | 0.6 | 1.2 | 1.0 | 5.0 | 4.3 | 4.5 | 1.3 |
| PocketFlow | 2.8 | 1.3 | 0.5 | 0.9 | 1.3 | 1.0 | 1.5 | 1.0 | 3.6 | 1.7 | 0.0 | 0.8 |
| TamGen | 1.5 | 2.1 | 1.0 | 0.7 | 1.2 | 0.0 | 0.8 | 1.5 | 1.4 | 4.0 | 7.0 | 1.5 |
| TargetDiff | 2.4 | 2.2 | 1.3 | 1.6 | 1.3 | 2.7 | 1.4 | 0.0 | 1.9 | 2.0 | 0.0 | 1.2 |

Table 8: Enrichment factors for each target based on Boltz-1 ipTM scores. *Indicates targets with a significant difference ($p < 0.05$) in Boltz-1 ipTM scores between active and random subsets of BindingDB (see Figure 8).

| | ESR1* | HCRTR1* | JAK1* | P2RX3* | KDM1A* | IDH1* | RIOK1 | NR4A1* | GRIK1* | FTO* | SPIN1 | CCR9* |
|---|---|---|---|---|---|---|---|---|---|---|---|---|
| BindingDB (active) | 19.9 | 14.7 | 13.1 | 10.3 | 2.9 | 2.0 | 1.3 | 2.1 | 11.3 | 4.3 | 0.0 | 11.3 |
| ProtoBind-Diff | 12.0 | 5.2 | 2.1 | 0.9 | 2.0 | 1.1 | 1.1 | 2.7 | 8.3 | 1.3 | 1.8 | 2.2 |
| REINVENT4 | 1.3 | 0.2 | 0.0 | 0.3 | 0.9 | 0.4 | 0.2 | 0.6 | 1.9 | 0.5 | 0.0 | 0.0 |
| Pocket2Mol | 12.2 | 0.4 | 0.7 | 1.2 | 0.6 | 1.9 | 0.9 | 2.5 | 6.6 | 4.4 | 0.5 | 4.9 |
| PocketFlow | 4.9 | 0.2 | 0.0 | 0.5 | 1.4 | 0.2 | 0.0 | 0.8 | 4.0 | 1.0 | 0.2 | 1.1 |
| TamGen | 0.2 | 0.3 | 0.1 | 0.7 | 1.2 | 0.6 | 0.2 | 0.7 | 1.4 | 3.1 | 0.2 | 1.1 |
| TargetDiff | 2.8 | 0.3 | 0.3 | 1.7 | 0.3 | 1.0 | 0.6 | 0.9 | 5.1 | 1.4 | 0.2 | 1.9 |

Table 9: Enrichment factors for each target based on Boltz-2 binary affinity probability scores. *Indicates targets with a significant difference ($p < 0.05$) in Boltz-2 scores between active and random subsets of BindingDB (see Figure 9).

