# OpenReview forum: "ProtoBind-Diff: Protein-Conditioned Discrete Diffusion for Structure-Free Ligand Generation"
_ICLR.cc/2026/Conference — Submitted to ICLR 2026_

### Official Review · Reviewer_rY6r · 2025-10-30

**Soundness:** 2
**Presentation:** 3
**Contribution:** 2
**Rating:** 2
**Confidence:** 3

**Summary:**

Authors propose a new masked diffusion language model to generate molecules with sequence-based representations of proteins and ligands.

**Strengths:**

Paper is well-written and easy to follow.
Comparisons are good. I appreciate the inclusion of very recent methods such as TamGen.

**Weaknesses:**

Doesn’t seem to be doing better than existing methods. ProtoBind only seems to do better on diversity and is on par with or worse than other methods on all other metrics, including vina docking score. Could the authors include error bars to show statistical significance of results?

Masked diffusion language modeling already exists. What is the methodological novelty in this method? Why limit to sequence data when structure data already exists?

**Questions:**

see weaknesses, above

---

> ### Author Response · Authors · 2025-11-22
>
> We sincerely thank you for your comments. We appreciate your positive feedback on the quality of the comparisons, the inclusion of recent models, and the clarity of the paper. We address your concerns below.
>
> > **“Could the authors include error bars to show statistical significance of results?”**
>
> We have included error bars in Table 1 and Table 2 in the revised manuscript.
>
> ---
>
> > **“ProtoBind only seems to do better on diversity and is on par with or worse than other methods on all other metrics, including vina docking score.”**
>
> Although our QED, SA, and validity scores are lower, our superior MMD results indicate that ProtoBind-Diff better reproduces the physicochemical property distribution of active molecules. We intentionally did not optimize for QED and SA, as these metrics do not necessarily correlate with binding affinity and can lead to artificial distributions. We acknowledge the validity score as a specific area for future improvement. Please also see our answers to Q5 and Q6 for Reviewer 44Qb for additional discussion.
>
> We explicitly state that our model does not achieve the best Vina docking scores: models such as PocketFlow and Pocket2Mol reach higher enrichment factors. Their training dataset, CrossDocked2020, was heavily augmented with Vina-generated poses, and we hypothesize that these models may be biased toward generating molecules that score well under Vina. This interpretation is supported by the observation that the EF of Pocket2Mol even exceeds that of the true active molecules. Furthermore, the relatively low EF observed for the true actives suggests that the Vina scoring function does not align well with actual binding activity. In addition, we evaluate our model using Boltz-1 and Boltz-2. As shown in the paper, ProtoBind-Diff achieves higher Boltz-1/2–derived enrichment factors than the baselines, suggesting that it generates ligands with higher genuine binding potential despite lower Vina scores.
>
> ---
>
> > **“Masked diffusion language modeling already exists.”**
>
> We agree that masked diffusion is an established method and has been applied to molecules before. However, to our knowledge, ProtoBind-Diff is the first target-aware text diffusion model conditioned directly on protein sequences via ESM-2 embeddings.
>
> ---
>
> > **“Why limit to sequence data when structure data already exists?”**
>
> We deliberately chose sequence conditioning to bypass the data bottleneck of structure-based methods. While structure-based models are limited to approximately 30,000 complexes in PDBbind, our sequence-based approach allowed us to train on over 1 million active pairs from BindingDB. Larger, more diverse datasets often lead to better generalization, and our main motivation for using sequence data was to unlock substantially larger training sets. Thus, we view sequence conditioning not as a limitation, but as a strategy that enables significantly larger training datasets and applicability to targets for which no high-quality crystal structure is available (or high-confidence predicted structures).
>
> We hope that this response addresses your questions and clarifies any remaining confusion.

---

### Official Review · Reviewer_y1TF · 2025-10-31

**Soundness:** 3
**Presentation:** 3
**Contribution:** 3
**Rating:** 6
**Confidence:** 4

**Summary:**

This paper introduces ProtoBind-Diff, a masked discrete diffusion model for structure-free ligand generation using protein sequence embeddings from the pretrained language model ESM-2. Unlike traditional methods relying on 3D structures, ProtoBind-Diff generates drug-like molecules for diverse protein targets, including those without known structures. Trained on over one million protein-ligand pairs from BindingDB, it outperforms or matches state-of-the-art 3D-structure-based and sequence-based models, demonstrating competitive binding affinity (notably with Boltz-1/2 predictors) and biologically interpretable attention patterns aligned with true binding residues.

**Strengths:**

* This work broadens applicability by generating ligands conditioned on protein sequences, eliminating the need for 3D structural data and enabling its use for targets with little to no structural information.
* This paper is well-written, with a clear presentation of both the motivation and the methodology.
* This paper provides a rich benchmark, evaluating 12 diverse targets from easy to hard classes and systematically compares against 3D generative models.
* The quantitative analysis of attention head distributions and their correlation with key protein binding sites is insightful.

**Weaknesses:**

* The architectural innovation of the model is quite limited, as it employs a standard cross-attention mechanism for condition injection, with the only distinction being the use of protein sequences as input.
* Although the idea for conditioning ligand generation on pure protein sequences is feasible, the authors need to further elaborate on why sequence-based conditioning offers advantages over 3D structure-based conditioning, especially in cases where real or predicted 3D structures are available.
* The paper lacks an in-depth discussion and experiments regarding the model architecture. For instance, it remains unclear how ProtoBind-Diff performs with varying parameter scales, different types of protein encoders, and alternative condition injection methods.
* The performance improvement of ProtoBind-Diff over the baseline models is minimal.

**Questions:**

* Could the authors provide a comparison with alternative condition injection methods? For example: 1) Pooling the protein sequence and injecting it using AdaLN; 2) Treating the protein sequence as the context for the ligand sequence and injecting by sharing q, k, v (similar to MM-DiT); 3) A hybrid approach combining both methods.
* Please provide a wider range of parameter scale combinations to validate the model's scalability.
* Please include ablations with a broader range of protein encoders, such as testing more advanced models like ESM-C to evaluate if they yield better results.
* One of my major concerns is whether replacing the protein encoder in ProtoBind-Diff with a structure-aware encoder of similar scale, such as SAProt-650M, while keeping the rest of the architecture unchanged, would result in significant performance differences. If the performance remains similar or is lower, it could support the authors' claim in the introduction regarding the limitations of using 3D structural inputs.
* Please include a comparison with baseline models in terms of parameter scale and inference speed.
* Please revisit the necessity of introducing the sequence-based condition in light of W.2 and Q.4.

---

> ### Author Response · Authors · 2025-11-22
>
> We sincerely appreciate your positive comments that the paper is well-written, that it broadens the applicability of the ligand generation for targets with little to no structural information, and that the analysis of attention heads is insightful. We address your concerns and questions below.
>
> > ‘Please include a comparison with baseline models in terms of parameter scale and inference speed.‘
>
> We compared the inference speed of all models on the same hardware (NVIDIA GeForce RTX 4070 Ti) with a batch size of 10. The results below are averaged over 3 runs generating 100 molecules total. We also provide the parameter scale of the models.
>
> | Model          | Latency (ms/mol) | Throughput (mol/s) | Parameter Count |
> | -------------- | ---------------- | ------------------ | --------------- |
> | REINVENT4      | 26.9 ± 0.0       | 37.21              | 5.81 M          |
> | TamGen         | 39.4 ± 0.6       | 25.36              | 112.12 M        |
> | Pocket2Mol     | 353.9 ± 12.4     | 2.83               | 3.71 M          |
> | ProtoBind-Diff | 657.7 ± 8.4      | 1.52               | 159.53 M        |
> | PocketFlow     | 1139.8 ± 26.7    | 0.88               | 0.22 M          |
> | TargetDiff     | 1683.5 ± 0.6     | 0.59               | 2.84 M          |
>
> Regarding condition injection and scaling: we agree that exploring alternative injection strategies (e.g., AdaLN or MM-DiT) and a wider range of parameter scales would give a more complete picture. However, retraining these variants would require substantial computational resources (and time) and is challenging for us to carry out within the rebuttal timeframe, so we do not include these ablations in the current work.
>
> Regarding Encoders (ESM-C / SAProt): we acknowledge that using structure-aware encoders like SAProt-650M or other protein encoders, such as ESM-C, is a promising direction. Our current focus was to validate the effectiveness of our proposed method with the current setup. Once again, given the extensive computational resources required to retrain these variations and the time constraints of the rebuttal period, we could not perform these specific large-scale ablations before this response. We ask you to revisit our answer next week, when we will try to finish the evaluations (most likely with SAProt-650M).

---

### Official Review · Reviewer_44Qb · 2025-11-01

**Soundness:** 3
**Presentation:** 2
**Contribution:** 2
**Rating:** 4
**Confidence:** 3

**Summary:**

This paper proposes a novel ligand generation model named ProtoBind-Diff, which uses the protein's amino acid sequence as a condition to guide molecule generation. The model is based on a masked discrete diffusion framework and is trained on the large-scale BindingDB dataset. Experiments demonstrate that ProtoBind-Diff generates ligands with chemical properties and binding affinity closely matching real active molecules.

**Strengths:**

- This paper proposes a "structure-free" paradigm for target-aware ligand generation
- By breaking free from 3D structure constraints, the model can be trained on BindingDB, which vastly exceeds the scale of traditional structural datasets.
- This paper conducts interpretability analysis, showing the models' capability to capture key binding residues even when not trained on 3D structures.

**Weaknesses:**

- The model does not resolve the same limitations it critiques in SBDD. The authors claim that SBDD models often ignore conformational flexibility and induced-fit effects, suffer from limited structural data, and may produce poor results for diversity and other key properties. However, the paper fails to demonstrate how the proposed sequence-based model addresses these same challenges.
- Poor conceptual and technical novelty. It doesn't discuss why sequence-based design would be preferable to structure-based design if structural data were available. The authors do not provide evidence of using sequence over structure, nor do they sufficiently argue against the strong and obvious alternative of using high-accuracy structure prediction tools (e.g., AlphaFold) or pocket prediction tools to generate inputs for established SBDD models. The core components, using a discrete diffusion model for molecular generation and a cross-attention mechanism to fuse conditioning information, are both well-established techniques.
- The evaluation framework is insufficient and potentially unreasonable. (As detailed in questions below).
- The paper lacks essential visualization results and case studies.

**Questions:**

- As for the potential advantage of this work in considering conformation flexibility and induced-fit effects, I recommend following evaluation works on related works (e.g., [1]). And it is recommended to discuss more about related works like this. [1] is a generative SBDD method that explicitly considers protein flexibility.
- During evaluation, how is the number of molecule atoms determined for baseline models? Is there any bias?
- Regarding the clustering method: Is it based on scaffold novelty? Or clustering based on other features?
- How the author ensure that the test set does not overlap with the training set? Was a target (sequence) similarity analysis performed?
- "Lower MMD values indicate greater similarity to the BindingDB reference set and thus better generation quality." How to explain this similarity regarding the goal of diversity?
- The author admits that validity cannot be optimized simultaneously with other molecular properties. But compared with SBDD models, where molecules are designed together with sequence and structure, this work only designs molecular sequence, yet still performs poorly in terms of validity. Could the authors explain the reason? The bad validity may indicate that the model does not learn the basic chemical rules.
- In Table 6, why are only MMD values reported instead of the real values/scores for the properties?
- Why is a low MMD (being closer to the distribution of known actives) considered better? For drug discovery, the goal is often to find novel candidates that surpass existing molecules (e.g., stronger competitive inhibitors or activators), not just reproduce the existing distribution.
- Have you tried fine-tuning the generative model by reinforcement learning [2] or preference optimization [3,4]? More discussion about this might further improve this submission.
- "All generative models performed well on targets where docking effectively distinguished active from inactive compounds, for example, ESR1, GRIK1, and CCR9." How is "effectively distinguished" formally defined? Is it defined by the Enrichment Factor?
- I understand the concern that Vina docking may have a high False Positive rate, as indicated by EF study in this paper, so I recommend the author to run other traditional evaluations for binding affinity, such as MMGBSA or MMPBSA.
- The attention-based analysis does not prove that the generated ligands actually form key interactions. Could the authors use PLIP [5] analysis to confirm if Non-Covalent Interactions were formed in accordance with the key residues and motifs?

**Note: I am willing to change my rating according to the rebuttal discussion and the paper revision.**

References:

[1] Integrating Protein Dynamics into Structure-Based Drug Design via Full-Atom Stochastic Flows. ICLR 2025

[2] Stabilizing policy gradients for stochastic differential equations via consistency with perturbation process, ICML 2024

[3] Antigen-Specific Antibody Design via Direct Energy-based Preference Optimization, NeurIPS 2024

[4] Decomposed Direct Preference Optimization for Structure-Based Drug Design, TMLR 2025

[5] PLIP: fully automated protein–ligand interaction profiler. Nucleic acids research, Nucleic Acids Research 2015

**Details Of Ethics Concerns:**

Only in silico evaluation was involved. No ethics concerns.

---

> ### Author Response · Authors · 2025-11-22
>
> **Response part 1**
>
> Thank you for your review. We address your questions below.
>
> ---
>
> > **1. [1] is a generative SBDD method that explicitly considers protein flexibility**
>
> We thank the reviewer for pointing us to DynamicFlow[1], a generative SBDD method that explicitly models protein flexibility and induced fit. We agree that the original version did not sufficiently discuss such approaches. In the revised manuscript we (i) described flexible SBDD methods including FlexSBDD and DynamicFlow in Related Works, (ii) revise the Introduction to emphasize that in recent approaches, such as DynamicFlow, pocket flexibility is addressed, and (iii) explicitly list extending our benchmark with these methods as future work. ProtoBind-Diff is a structure-free ligand generator: it conditions on protein sequence embeddings and outputs SMILES, but does not generate protein pocket conformations or apo→holo transitions. Consequently, the induced-fit evaluation protocols of [1] are not directly applicable in our setting.
>
>
> ---
>
> > **2. During evaluation, how is the number of molecule atoms determined for baseline models? Is there any bias?**
>
> During evaluation we did not manually fix or tune the number of atoms for any baseline; we used the authors’ default settings in each case.
>
> * TargetDiff: `sample_num_atoms = prior`, where the atom count is sampled from the pocket-dependent prior in the original code.
> * Pocket2Mol: autoregressive generation until the model predicts termination; no extra atom-count control.
> * PocketFlow: we kept the default `max_atom_num = 35` from the official implementation’s command example.
> * REINVENT4 and TamGen: these are SMILES language models that generate until an end-of-sequence token; we did not add any atom-count hyperparameter beyond their standard maximum sequence length.
>
> Thus we did not introduce any additional constraints on molecule size; any differences in atom counts come from the models themselves, not from our evaluation protocol.
>
> > **3. Regarding the clustering method: Is it based on scaffold novelty? Or clustering based on other features?**
>
> Our clustering is not scaffold-based. We cluster molecules using binary topological fingerprints and Tanimoto similarity. Concretely, for each molecule we compute a fixed-length binary molecular fingerprint (RDKit Morgan fingerprints, radius 2, 2048 bits), which encodes the presence of local substructures. Tanimoto similarity between two fingerprints measures the overlap of these substructures (1 = identical, 0 = no shared features). We then apply the Taylor–Butina clustering algorithm with Tanimoto similarity as the distance measure and a cutoff of 0.2. The “Diversity” metric reported in Table 1 and Table 2  is defined as the number of resulting clusters divided by the total number of generated molecules. The same fingerprint-based clustering is also used in our training-time resampling scheme.
>
> > **4. How the author ensure that the test set does not overlap with the training set? Was a target (sequence) similarity analysis performed?**
>
> ProtoBind-Diff is trained on the full cleaned BindingDB dataset; the 12 benchmark targets are not held out. Our goal was to compare the performance of our model to the baselines, which were trained oт different datasets. It is not trivial to find targets that are both not included in the training sets of the selected baselines and have a reasonable number of active protein-ligand pairs. We also did not want to create new train test splits for each model and retrain them. Thus, we decided to compare the performance of each model on ‘easy’ and ‘hard’ targets. The logic here is that if a target has few data points in the BindingDB, in all other datasets it is also covered poorly, since BindingDB is the largest, most complete open source dataset of protein ligand pairs.
> Protein sequences were clustered using CD-HIT at 60 % identity. For each cluster we counted how many protein–ligand pairs appear in the training data, and then selected 6 “easy” targets from clusters with many examples (> 4000) and 6 “hard” targets from clusters with few examples (< 350) (Table 4).
> In drug–target interaction classification and binding affinity regression, it is important to control for ligand-based data leakage [2], as performance can be artificially inflated when highly similar ligands appear in both training and test sets. In the generative setting, where only the protein is provided at inference time, this type of ligand-based leakage does not arise. Moreover, we evaluate all methods only on novel molecules (Tanimoto < 0.5 to active BindingDB ligands for a given target), so our performance claims are not driven by ligand-based bias.

---

> > ### Author Response · Authors · 2025-11-22
> >
> > **Response part 2**
> >
> > > **5. "Lower MMD values indicate greater similarity to the BindingDB reference set and thus better generation quality." How to explain this similarity regarding the goal of diversity?**
> >
> > Diversity of a set of molecules is quantified as the number of clusters divided by the total number of molecules, where clusters are defined based on molecular fingerprints as described in our response to Q4. This metric captures how structurally spread out the generated molecules are.
> > By contrast, MMD measures how close the distribution of basic physicochemical properties (e.g., LogP, molecular weight, number of rotatable bonds, etc.) is to that of the BindingDB reference set. In other words, lower MMD means that the generated molecules occupy a similar region of physicochemical property space as BindingDB, not that they are structurally redundant. Because chemical space is extremely large, it is entirely possible to match the reference property distribution while still generating structurally diverse molecules.
> >
> > ---
> >
> > > **6. But compared with SBDD models, where molecules are designed together with sequence and structure, this work only designs molecular sequence, yet still performs poorly in terms of validity. Could the authors explain the reason?**
> >
> > Validity as a metric can be trivially increased by generating very short and simple molecules, which typically leads to low uniqueness and diversity. For this reason, we did not explicitly optimize ProtoBind-Diff for validity alone: in our view, it is preferable to filter out a moderate number of invalid designs than to obtain a large number of trivial but chemically uninteresting molecules. In addition, our model is trained in the SMILES space, where small token permutations often render a molecule invalid, making syntactic validity hard to achieve. We acknowledge this limitation of study design and plan to explore more robust sequence representations such as SELFIES in future work to improve validity without sacrificing diversity.
> >
> > ---
> >
> > > **7. In Table 6, why are only MMD values reported instead of the real values/scores for the properties?**
> >
> > We use MMD here because the goal is to compare full distributions of molecular properties between generated molecules and the BindingDB reference set. For each property, Table 6 summarizes this distributional difference as a single scalar (MMD) averaged over the 12 targets, which keeps the main text compact. MMD is more informative for distribution comparison than raw means, which can coincide even when the underlying distributions differ. Figure 6 shows the mean property values for each target and each generative model.
> >
> > ---
> >
> > > **8. Why is a low MMD (being closer to the distribution of known actives) considered better?**
> >
> > We agree with the reviewer that the ultimate goal in drug discovery is to find novel, more potent compounds. For a given target, known actives occupy a fairly restricted region of physicochemical space (MW, logP, HBD/HBA counts, charge, etc.), because the geometry, charge,HBD/HBA of the pocket determine the subspace of molecules that would successfully fit inside. If a generator drifts far outside this distribution, it is likely to produce poor drug candidates.
> > Distribution-matching metrics are also standard in de novo design benchmarks. GuacaMol [3] compares models to the reference set via KL divergence over physicochemical property distributions (e.g., MW, logP, HBD/HBA counts, ring counts) and Fréchet ChemNet Distance (FCD), while MOSES [4] reports Wasserstein distances between generated and reference distributions of MW, logP, QED, SA, along with FCD, in addition to novelty and goal-directed tasks. Likewise, pocket-based models such as Pocket2Mol and subsequent TargeDiff compare ratios of rings of different sizes, and structural characteristics (bond lengths, angles)  between generated molecules and test set.
> > Thus, low MMD indicates that, conditional on a target, the generated set lies in a chemically plausible region similar to known actives; the Tanimoto similarity, however, can be low. The goal of “surpassing” existing molecules is addressed in our binding-affinity analysis (Boltz and Vina scores).
> >
> > ---

---

> > > ### Author Response · Authors · 2025-11-22
> > >
> > > **Response part 3**
> > >
> > > > **9. Have you tried fine-tuning the generative model by reinforcement learning [2] or preference optimization [3,4]? More discussion about this might further improve this submission.**
> > >
> > > We have not applied reinforcement learning or preference optimization to ProtoBind-Diff in this work. We agree that combining ProtoBind-Diff with RL or preference optimization is a natural next step. In particular, integrating Boltz-2 as a reward / preference model in the spirit of recent methods [5–7] could be used to steer generation toward higher-affinity regions of chemical space while using ProtoBind-Diff as the underlying prior. We are currently exploring such extensions, but the experiments are not mature enough to report and are therefore left for future work. We have added a brief discussion of this direction to the Conclusion / Future Work section.
> > >
> > >
> > > ---
> > >
> > > > **10. How is "effectively distinguished" formally defined? Is it defined by the Enrichment Factor?**
> > >
> > > Yes, by ‘effectively distinguished’ we refer to targets where docking scores for known active molecules and inactive molecules are clearly separated. In practice, this means (i) the mean docking score for active ones is significantly better than for inactive molecules, as confirmed by a statistical test (p-value  0.05), and (ii) this separation results in high enrichment factors (EF).
> > >
> > > ---
> > >
> > > > **11. I understand the concern that Vina docking may have a high False Positive rate, as indicated by EF study in this paper, so I recommend the author to run other traditional evaluations for binding affinity, such as MMGBSA or MMPBSA**
> > >
> > > Thank you for this comment. We agree that MMGBSA/MMPBSA would provide a more physically grounded estimate of binding affinity, but both these methods are considerably more computationally expensive than docking. Applying them systematically to 12 targets and multiple generative models would require a considerable time which is quite limited during the rebuttal stage.
> > >
> > > ---
> > >
> > > > **12. Could the authors use PLIP [5] analysis to confirm if Non-Covalent Interactions were formed in accordance with the key residues and motifs?**
> > >
> > > Our model is trained purely on sequence and does not predict 3D structures. Running PLIP would therefore mainly probe Boltz-1’s structural prior rather than provide ground-truth evidence that ProtoBind-Diff found specific non-covalent interactions. The CCR9 predicted complex is shown mostly for illustrative purposes. Our goal in the attention analysis was to check whether the model systematically focuses on known binding-site regions (BioLiP residues aggregated in a 3-residue window), not to claim that attention peaks coincide with individual H-bonds, salt bridges, or π–stacking contacts. Following the reviewer’s suggestion, we used PLIP to investigate the CCR9 case and observed that among the residues highlighted by attention, only Y126 forms an H-bond with atom N16 on the tetrazole ring, and a hydrophobic interaction with atom C13 on the benzole ring. According to PLIP, A306-F308 do not form Non-Covalent interactions.
> > >
> > > ---
> > >  **Literature**
> > >
> > > 1. Zhou, X., Xiao, Y., Lin, H., He, X., Guan, J., Wang, Y., ... & Ma, J. (2025). *Integrating Protein Dynamics into Structure-Based Drug Design via Full-Atom Stochastic Flows.* arXiv preprint arXiv:2503.03989.
> > > 2. Durant, G., Boyles, F., Birchall, K., Marsden, B., & Deane, C. M. (2025). *Robustly interrogating machine learning-based scoring functions: what are they learning?* Bioinformatics, 41(2), btaf040.
> > > 3. Brown, N., Fiscato, M., Segler, M. H., & Vaucher, A. C. (2019). *GuacaMol: benchmarking models for de novo molecular design.* Journal of chemical information and modeling, 59(3), 1096–1108.
> > > 4. Polykovskiy, D., Zhebrak, A., Sanchez-Lengeling, B., Golovanov, S., Tatanov, O., Belyaev, S., ... & Zhavoronkov, A. (2020). *Molecular sets (MOSES): a benchmarking platform for molecular generation models.* Frontiers in pharmacology, 11, 565644.
> > > 5. Zhou, X., Wang, L., & Zhou, Y. (2024). *Stabilizing policy gradients for stochastic differential equations via consistency with perturbation process.* arXiv preprint arXiv:2403.04154.
> > > 6. Zhou, X., Xue, D., Chen, R., Zheng, Z., Wang, L., & Gu, Q. (2024). *Antigen-specific antibody design via direct energy-based preference optimization.* Advances in Neural Information Processing Systems, 37, 120861–120891.
> > > 7. Cheng, X., Zhou, X., Yang, Y., Bao, Y., & Gu, Q. (2024). *Decomposed direct preference optimization for structure-based drug design.* arXiv preprint arXiv:2407.13981.

---

### Official Review · Reviewer_Pprt · 2025-11-01

**Soundness:** 2
**Presentation:** 2
**Contribution:** 2
**Rating:** 2
**Confidence:** 3

**Summary:**

The paper proposes ProtoBind-Diff, a structure-free masked discrete diffusion model that conditions ligand generation on protein sequence embeddings (ESM-2) via cross-attention. Trained on >1M protein–ligand activity pairs, the method aims to avoid dependencies on 3D pocket structures. The authors report: (i) molecular property distributions close to actives, (ii) favorable enrichment under Boltz-1 on a 12-target benchmark, and (iii) attention maps aligning with binding-site residues, suggesting biologically reasonable priors learned from sequence alone. Pocket-free generation is positioned as a scalable pathway for orphan/rapidly emerging targets where 3D structures are unavailable.

**Strengths:**

1. **Clear problem motivation & scope**: Removing the dependency on 3D pocket structures addresses a real deployment bottleneck (limited and biased complex structures in PDBbind compared to activity-centric resources like BindingDB).
2. **Competitive performance vs 3D baselines**: Empirical comparisons against Pocket2Mol and PocketFlow (flow-matching–based SBDD) support the claim that sequence-conditioned diffusion can rival pocket-conditioned methods on multiple targets.

**Weaknesses:**

1. **Heavy reliance on proxy metrics; risk of bias**
   - The core claims depend on Boltz-1/2 and Vina. While Boltz-1/2 are strong modern predictors, their training distribution and the overlap with commonly used ligand/target corpora can confound absolute gains. Please (a) report distributional overlap checks (ligand scaffolds and protein family homology) against your training sources; (b) complement with leak-controlled external tests such as LP-PDBbind or other curated splits explicitly designed to reduce leakage.
2. **Comparisons to target-aware sequence models need tighter controls**
   - TamGen is a natural, strong comparison for target-aware chemical LMs. Please ensure identical novelty filters, deduplication, sampling budgets, and stratified reporting on “easy vs. low-data” targets; add statistical tests (e.g., stratified bootstrap CIs) on key metrics.
3. **Validity/uniqueness vs. quality trade-offs**
   - Provide sampling curves vs. step count, mask rate, and nucleus-p, reporting Validity, Uniqueness, Novelty, Property MMD, and Boltz-1 to show practical operating points.
4. **Compute & scalability reporting**
   - Since the paper’s value proposition includes scalability without 3D pockets, please add throughput/latency/memory comparisons to Pocket2Mol/PocketFlow/TamGen at equal batch sizes and similar sampling budgets.

**Questions:**

1. **Leak-control protocol**: How do you guard against scaffold-level and protein-family leakage between BindingDB training and your 12-target test set? Please include a similarity matrix and thresholding policy, then recompute headline metrics after removing near-neighbors.
2. **Low-data targets**: Which components (ESM-2 conditioning vs. resampling vs. augmentation) drive robustness on sparse targets? Please include ablation tables.
3. **Generalization beyond BindingDB**: Any early results on curated, leak-controlled external testbeds (e.g., LP-PDBbind) or other activity datasets to support broader claims?
4. **Contextualization of baselines**: Briefly summarize Pocket2Mol (3D pocket-conditioned autoregressive assembly) and PocketFlow (flow-matching with interaction priors) to help readers who are less familiar with SBDD baselines appreciate where sequence-only diffusion stands.

---

> ### Author Response · Authors · 2025-11-22
>
> 1. Leak-control protocol
>
> To guard against scaffold-level and protein-family leakage between BindingDB training and the 12-target test set, we applied both sequence and chemical similarity filtering.
> All BindingDB protein sequences were clustered using CD-HIT at 60% sequence identity, and we quantified for each cluster how many protein–ligand pairs appeared in the training data. We then selected 6 “easy” targets from clusters with abundant training representation (> 4000 pairs) and 6 “hard” targets from clusters with scarce representation (< 350 pairs) (see Table 4). This clustering defines our protein-family similarity matrix and serves as the thresholding policy for controlling target-level leakage.
> For ligands, we computed pairwise Tanimoto similarities between test-set ligands and all training ligands. Molecules with Tanimoto ≥ 0.5 were treated as near neighbors and excluded from the evaluation. Headline metrics were recomputed after removing these neighbors, confirming that relative performance trends remained consistent.
> Although ProtoBind-Diff was trained on the full cleaned BindingDB dataset (without holding out the 12 benchmark targets), the evaluation explicitly distinguishes “easy” (well-represented) and “hard” (under-represented) clusters to approximate out-of-distribution generalization. Since our generative model conditions only on protein sequences and generates novel ligands (Tanimoto < 0.5 to any training molecule), ligand-level leakage is further minimized.
>
> 2. Low-data targets and Generalization beyond BindingDB
>
> To further assess generalization under leak-controlled conditions, we evaluated ProtoBind-Diff and its ablations on a subset of LP-PDBbind. We first clustered proteins from LP-PDBbind together with targets from our training set using CD-HIT at 60 % sequence identity to enforce sparse coverage and selected five representative novel targets (3l7a, 5abh, 5edu, 3dnt, 3pn4). All models were evaluated using the same inference and metric pipeline as in BindingDB experiments. The presence of resampling and augmentation preserved higher molecule validity and chemical quality metrics (QED, SAScore). Because LP-PDBbind contains very few experimentally confirmed active ligands for chosen target, Boltz-1/2 and Vina scores could not be reliably used to assess whether these methods distinguish active from inactive molecules; however, the consistent trends across metrics and the strict sequence and scaffold filtering (Tanimoto < 0.5) confirm that ProtoBind-Diff generalizes beyond BindingDB and remains robust to protein and data sparsity.
>
> | Model                | ESM-2 model |    Validity |         QED |     SAScore |          MolWt |     MolLogP | NumRotatableBonds |       CalcTPSA |   RingCount | Uniqueness |
> | -------------------- | ----------: | ----------: | ----------: | ----------: | -------------: | ----------: | ----------------: | -------------: | ----------: | ---------: |
> | No rand. + no sampl. |        650M | 0.72 ± 0.03 | 0.43 ± 0.04 | 3.09 ± 0.16 |  432.50 ± 3.77 | 2.32 ± 0.52 |       8.54 ± 0.43 |  123.97 ± 5.54 | 2.58 ± 0.22 |       0.99 |
> | No rand. + sampl.    |        650M | 0.70 ± 0.01 | 0.38 ± 0.01 | 3.85 ± 0.04 | 649.10 ± 10.14 | 2.74 ± 0.05 |      12.03 ± 0.28 |  171.51 ± 3.96 | 3.96 ± 0.03 |       1.00 |
> | Rand. + sampl.       |        650M | 0.80 ± 0.03 | 0.41 ± 0.03 | 3.37 ± 0.06 | 456.27 ± 48.80 | 1.99 ± 0.46 |       8.91 ± 0.95 | 138.57 ± 12.72 | 2.66 ± 0.37 |       1.00 |
> | Rand. + sampl.       |        150M | 0.50 ± 0.01 | 0.48 ± 0.01 | 3.41 ± 0.02 |  554.33 ± 4.44 | 3.12 ± 0.05 |       9.73 ± 0.10 |  136.50 ± 2.22 | 3.92 ± 0.03 |       1.00 |
>
> 3. Contextualization of baselines
>
> We thank the reviewer for this suggestion. In the Related Work section, one can already find a brief overview of Pocket2Mol and PocketFlow alongside other structure-based molecular generation approaches. However, we agree that this discussion can be further enriched to help readers less familiar with SBDD baselines appreciate the distinction from sequence-only diffusion. We have slightly extended the description of Pocket2Mol and PocketFlow in the Related works section. Specifically, Pocket2Mol employs an E(3)-equivariant autoregressive assembly that incrementally places atoms inside a 3D protein pocket using geometric and bonding cues, while PocketFlow adopts a flow-matching paradigm with interaction priors (e.g., distance and pharmacophore constraints) to generate physically consistent ligands within pocket coordinates.
>
> 4. Compute & scalability reporting
>
> Please look at the answer to Reviewer y1TF, where we answer related question.

---

> > ### Comment · Reviewer_Pprt · 2025-11-26
> >
> > Thanks for the update. I think the major problems posted in my review have been addressed. I have raised my score.

---

### Meta-Review · Area_Chair_AdH8 · 2026-01-06

**Summary:**

The paper proposes ProtoBind-Diff, a structure-free masked discrete diffusion model designed for protein-conditioned ligand generation. By leveraging pre-trained ESM-2 protein sequence embeddings and utilizing cross-attention mechanisms, the method aims to generate specific ligands without relying on 3D structural data of protein-ligand complexes. The authors train the model on a large-scale dataset (BindingDB) comprising over one million active pairs. The study evaluates the model against several structure-based baselines on a 12-target benchmark, claiming competitive performance in generating chemically valid and novel molecules.

Strengths:
(1) This work broadens applicability by generating ligands conditioned on protein sequences, eliminating the need for 3D structural data and enabling its use for targets with little to no structural information.
(2) This paper provides a rich benchmark, evaluating 12 diverse targets from easy to hard classes and systematically compares against 3D generative models.
(3) This paper conducts interpretability analysis, showing the models' capability to capture key binding residues even when not trained on 3D structures.

Weakness:
(1) Doesn’t seem to be doing better than existing methods. ProtoBind only seems to do better on diversity and is on par with or worse than other methods on all other metrics, including vina docking score.
(2) The architectural innovation of the model is quite limited, as it employs a standard cross-attention mechanism for condition injection, with the only distinction being the use of protein sequences as input.
(3) The evaluation framework is insufficient and potentially unreasonable.

**Reviewer Concerns:**

Upon a careful examination of the manuscript, the reviews, and the authors' rebuttal, I observe that while the authors addressed some concerns, several critical issues remain unaddressed or were dismissed during the rebuttal phase. Specifically, the authors failed to provide a direct response or experimental data regarding the need for tighter controls in comparisons with target-aware sequence models (e.g., ensuring identical novelty filters, deduplication, and sampling budgets for the TamGen baseline). Furthermore, requests for a detailed analysis of validity/uniqueness versus quality trade-offs—such as providing sampling curves for step count, mask rate, and nucleus-p to demonstrate practical operating points—were not adequately addressed in the response text.

Crucially, the recommendation to validate binding affinity using more robust physics-based evaluations like MMGBSA or MMPBSA—to mitigate the high false-positive risks associated with Vina docking found in the EF study—was declined due to computational costs. The absence of these validations significantly weakens the claims regarding the biological relevance of the generated ligands.

Finally, Reviewers 44Qb, y1TF, and rY6r expressed significant concerns regarding the architectural novelty of the proposed method, with Reviewers y1TF and rY6r noting that the empirical improvements over existing baselines appear limited. The authors' rebuttal did not sufficiently resolve these concerns regarding the incremental nature of the technical contribution.

**Reviewer Scores:**

The paper received mixed initial feedback, consisting of one positive review and three negative reviews (Pprt (2), 44Qb (4), y1TF (6), and rY6r (2)). While reviewer Pprt expressed a willingness to increase their score following the authors' rebuttal, a consensus among the reviewers has not been reached.

In conclusion, I believe this manuscript requires substantial revision and more rigorous experimental validation to meet the acceptance standards of ICLR. Therefore, I recommend rejection at this time.

---

### Decision · Program_Chairs · 2026-01-26

Reject